# Resin acids play key roles in shaping microbial communities during degradation of spruce bark

Amanda Sörensen Ristinmaa [1], Albert Tafur Rangel [1,2], Alexander Idström[3], Sebastian Valenzuela [4], Eduard J. Kerkhoven [1,2], Phillip B. Pope [5,6], Merima Hasani[3,7] & Johan Larsbrink [1,7] ✉

The bark is the outermost defense of trees against microbial attack, largely thanks to toxicity and prevalence of extractive compounds. Nevertheless, bark decomposes in nature, though by which species and mechanisms remains unknown. Here, we have followed the development of microbial enrichments growing on spruce bark over six months, by monitoring both chemical changes in the material and performing community and metagenomic analyses. Carbohydrate metabolism was unexpectedly limited, and instead a key activity was metabolism of extractives. Resin acid degradation was principally linked to community diversification with specific bacteria revealed to dominate the process. Metagenome-guided isolation facilitated the recovery of the dominant enrichment strain in pure culture, which represents a new species (*Pseudomonas abieticivorans* sp. nov.), that can grow on resin acids as a sole carbon source. Our results illuminate key stages in degradation of an abundant renewable resource, and how defensive extractive compounds have major roles in shaping microbiomes.

Bark is the outer protective barrier of trees and shields them from abiotic and biotic stress, both thanks to gradual peeling during growth and its chemical composition. The bark structure differs from wood in that it typically contains a higher amount of both ash (-5%) and so-called extractive compounds (extractives, -20–30%), in addition to the regular wood polymers cellulose, hemicelluloses, and lignin[1–3]. Extractives are secondary metabolites with diverse properties that can be extracted using hydrophilic or hydrophobic solvents. Compared to wood, the chemical composition of bark remains less well understood and it also differs widely among tree species, especially regarding the extractives, and may also differ depending on season and sampling point[4]. Spruce is one of the dominant tree species in northern

hemisphere forests and of high economic value, and its bark comprises around 10–15% of the volume at harvest. In the Nordic countries alone, bark is produced in close to 400 million $m^3$ per year[5], but it is largely seen as an industrial side stream and is either left to rot in the forest or burnt for energy at the mill. In addition to wood polymers, spruce bark contains a high amount of lipophilic extractives, mainly resin acids (>10 mg/g dry bark), triglycerides (-8 mg/g), steryl esters (-5 mg/g), fatty acids (-2.5 mg/g), and sterols (-1.3 mg/g) (Fig. 1)[3].

Extractives generally inhibit microbial attack through cellular toxicity or by precipitating proteins, and this is also true for spruce extractives[6,7]. Resin acids in particular are known to be toxic to microorganisms[7,8] as well as water-living animals with $LC_{50}$ values

[1]Department of Life Sciences, Chalmers University of Technology, SE-412 96 Gothenburg, Sweden. [2]Novo Nordisk Foundation Center for Biosustainability, Technical University of Denmark, DK-2800 Kgs Lyngby, Denmark. [3]Department of Chemistry and Chemical Engineering, Chalmers University of Technology, SE-412 96 Gothenburg, Sweden. [4]Department of Medical Biochemistry and Cell Biology, University of Gothenburg, SE-405 30 Gothenburg, Sweden. [5]Faculty of Biosciences, Norwegian University of Life Sciences, NO-1433 Ås, Norway. [6]Faculty of Chemistry, Biotechnology and Food Science, Norwegian University of Life Sciences, NO-1433 Ås, Norway. [7]Wallenberg Wood Science Center, Chalmers University of Technology, SE-412 96 Gothenburg, Sweden. ✉e-mail: johan.larsbrink@chalmers.se

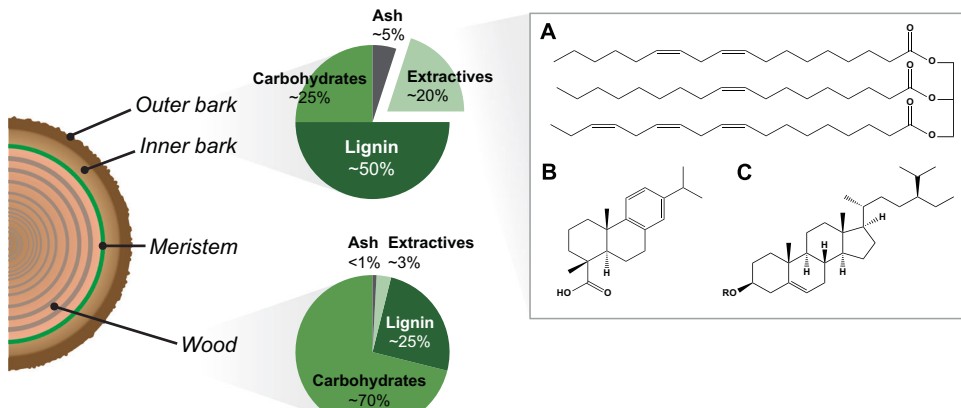

**Fig. 1 | Overview of the structure of tree bark compared to wood.** The bark can be classified as inner or outer, which depends on its proximity to the external environment and is governed to a large extent by its age. Compared to regular wood, the bark is enriched in minerals (ash) as well as the class of molecules referred to as extractives, which have highly variable properties. In the box, representative lipophilic extractive groups in spruce bark are shown: **A** triglycerides (glycerol linked to the three unsaturated fatty acids octadecenoic acid (C18:1); octadecadienoic acid (C18:2); octadecatrienoic acid (C18:3), **B** resin acids (dehydroabietic acid), and **C** sterols (R=H, β-sitosterol) and steryl esters (R=alkyl chain, e.g., octadecanoic acid).

(concentration lethal to half of a population) of sub-mg/L[9]. Despite the antimicrobial properties of bark, it is degraded in nature over time, but compared to the extensive literature on the biological degradation of wood, data on the molecular details of microbial bark degradation are sparse. Snapshot studies of which microbial communities can colonize bark of different trees have been conducted, for instance, analyzing bark remaining on logs or samples buried in soil after years of incubation[10–12]. However, how these communities correlate to degradation stage is currently unclear, as simultaneous detailed chemical analyses of the bark have not been conducted over time.

Regarding the degradation of individual spruce extractives, there are reports of non-toxic triglycerides and fatty acids, as well as toxic resin acids. Triglycerides have been shown to be degraded by the fungi *Phlebiopsis gigantea*, *Trametes versicolor*, and *Bjerkandera spp*[13,14]. Additionally, sterols and unsaturated fatty acids were degraded by *Phanerochaete velutina* and *Stropharia rugosoannulata*[15], and free and esterified sterols by *Phlebia radiata* and *Poria subvermispora*[16]. Bacteria have also been shown to degrade triglycerides, sterols, and resin acids[17]. More detailed investigations of resin acid degradation by *Pseudomonas* and *Paraburkholderia* suggested a degradation pathway of abietane-type resin acids[18,19]. However, it is presently unclear whether certain extractive groups have a larger effect on microbial communities and their development during bark degradation, and whether individual microorganisms tolerate, metabolize, and/or detoxify bark extractives. The composition and development of microbial communities during active decomposition of bark cannot be directly compared to similar studies of wood degradation, due to the large proportion of toxic extractive compounds in the bark material[20,21].

In order to understand how bark can be degraded, the process needs to be monitored over time and both microbial and chemical analyses need to be combined, to answer the questions of what type of species may be involved or dominate and what bark compounds are mainly affected. Here, we have investigated spruce bark degradation over 6 months using an enriched microbial community sourced from an industrial bark pile, and combined several methods to obtain a broad picture of the entire process. The enrichment development was studied by marker gene and metagenomics sequencing, and the simultaneous chemical changes of the carbohydrates, lignin, and extractives were monitored. We identified resin acids as key molecules to be degraded to allow the expansion of microbial diversity, while carbohydrates were metabolized to a lesser extent. Bacteria rather than fungi were linked to resin acid degradation, and we successfully isolated and genome-sequenced a strain of the main resin acid-

degrading bacterium, representing the new species *Pseudomonas abieticivorans*. We confirmed that *P. abieticivorans* can grow on resin acids as the sole carbon source, which is attributed to a large gene cluster encoding this dedicated function. Our holistic study paves the way for detailed understanding of microbial degradation of bark, which is important information concerning both the overall carbon cycle in forests and future development of valorization strategies for complex renewable bark biomass.

## Results

Bark was sourced from a pulp and paper mill after having been stripped from the logs. To sterilize the bark without excessively disrupting its structure, it was gamma-irradiated instead of autoclaved, as elevated temperatures and pressure would release or modify the extractives, and doses over 20 kGy were found to be enough to stop any growth. An active inoculum was prepared from unsterilized bark, and separate cultures were inoculated and collected over 24 weeks, with one fraction per sample analyzed by DNA extraction, and one fraction analyzed chemically.

### Chemical analyses of microbial spruce bark degradation

To catalog the organic and inorganic landscapes of the bark throughout the 6-month cultivation cycle, we first analyzed the major components of the bark: carbohydrates, lignin, extractives, and ash. In uninoculated (abiotic/blank) bark samples, very little differences could be observed overall throughout the process. The ash content, however, appeared to increase by ~53% in the biotic samples over the entire cultivation (Fig. S1), which we ascribe to loss of $CO_2$ or other volatiles. As a portion of each sample was sent for DNA extraction and sequencing, and the ash cannot disappear from the material, the proportional ash content was used to normalize the data in the biotic samples. The overall mass loss represents 3.6 g per 10 g sample, but the total bark consumption should be considered higher as microbial cells also represent a fraction of the total carbon. The bark carbohydrates were expected to be a major nutrient source for the consortium, and the monosaccharide composition was determined. Surprisingly, there were only small decreases in hemicellulose-derived monosaccharides (mannose, galactose, arabinose, xylose) (Figs. S1, S2). However, a larger effect could be seen for glucose (Figs. S1, S2), where a slow but continuous decrease could be observed which could be attributed to cellulose- or perhaps more likely starch degradation, as spruce bark can contain significant amounts of this more accessible polysaccharide[4]. In the blank sample, an apparent increase in glucose

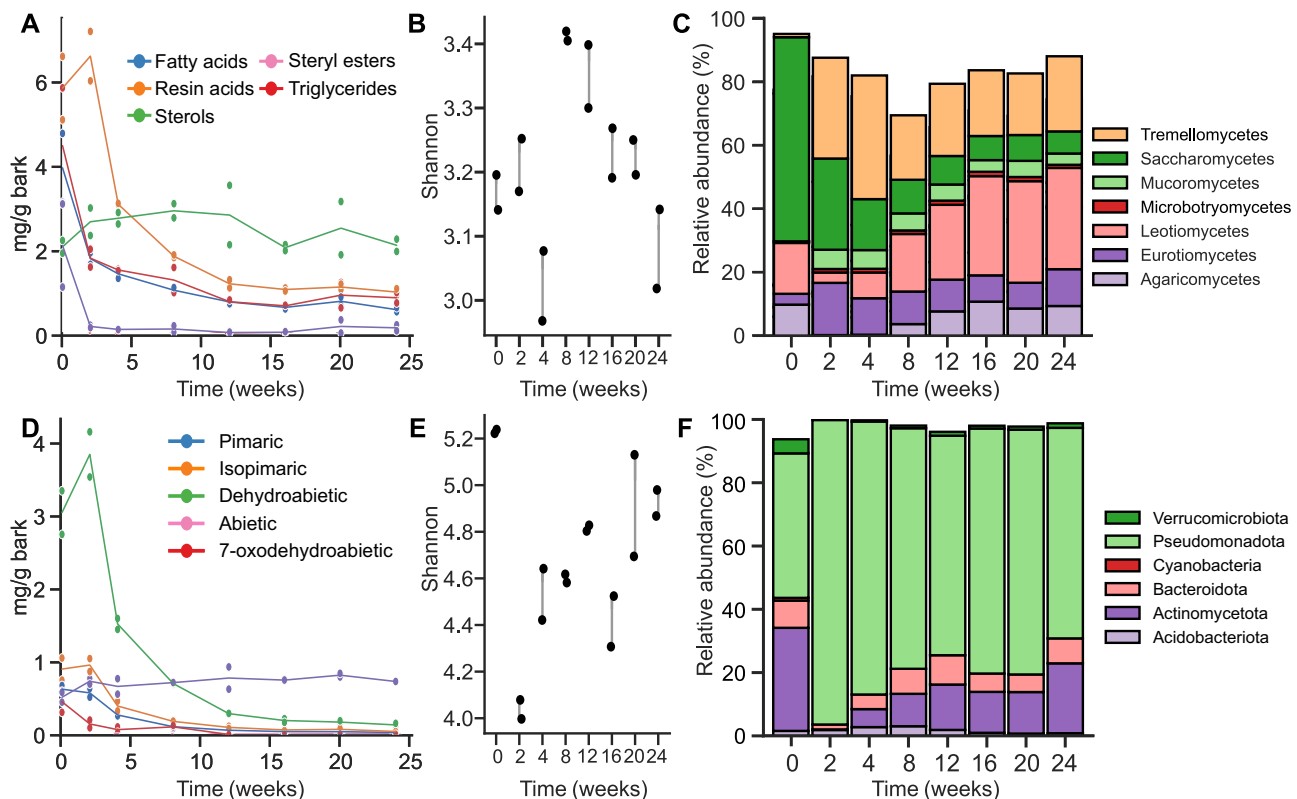

**Fig. 2 | Effect of inoculating sterile bark with a culture containing a mixture of both bacteria and fungi. A** Lipophilic extractive groups in the acetone extract over time. The resin acid concentration decreased after two weeks. **D** Individual resin acid compounds, which all decreased over time except 7-oxodehydroabietic acid. Alpha diversity of the (**B**) fungal community (class), and (**E**) bacterial community (phylum) over time, expressed as the Shannon diversity index, with data points from the same week linked by a line for clarity. **C** Relative abundance of fungal classes and (**F**) relative abundance of bacterial phyla, over time and derived from two biological replicate experiments. Individual data points are displayed in (**A**) and (**D**) with lines showing the means.

was observed in the 24-week sample, possibly stemming from less stable glycoconjugates than polysaccharides. Lignin degradation is generally a slow process performed by white-rot fungi, and potential degradation was measured by monitoring acid-insoluble and acid-soluble lignin, which showed no obvious changes either in biotic or abiotic samples (Figs. S1, S2). These collected results imply that the largest effect of microbial degradation could be found in the extractives.

The bark was extracted using acetone, to enable broad recovery of different extractive types, and the extractive yield of the untreated sample was similar to previous studies using acetone (Figs. S1, S2)[4]. As hypothesized, the amount of extractives decreased significantly over time for the biotic sample with 46% decrease in extractive dry weight after 2 weeks and 76% decrease after 24 weeks. In the abiotic sample no significant differences were observed (Fig. S2), indicating that bark extractives are a major carbon source during initial metabolism of spruce bark by the consortium.

**Extractive analysis reveals microbial degradation of resin acids**
To investigate which extractive types were mainly degraded, they were identified and quantified using gas chromatography (GC-MS and GC-FID, Table S1), and further characterized using 2D-NMR. As mentioned, resin acids are major lipophilic extractives in spruce bark, predominantly dehydroabietic acid, abietic acid, isopimaric acid, and 7-oxodehydroabietic acid, and these are regarded as highly toxic to microorganisms[6,7]. Interestingly, we observed drastic decreases in the resin acids, especially during initial growth, apart from 7-oxodehydroabietic acid which was rather stable throughout the cultivation. 7-Oxodehydroabietic acid is, however, a known degradation intermediate[19], and our results suggest continuous resin acid

degradation (Fig. 2). The apparent increase in dehydroabietic acid in the first time point may suggest release from the bark matrix, which we cannot explain since resin acids are not known to be linked to other bark molecules. However, the observation may point to some of the resin acids being, for instance, weakly bound to the bark matrix or abietic acid being microbially transformed into dehydroabietic acid during the cultivation[22]. Resin acids are known to be significantly affected during wood storage, though no changes were observed in the abiotic control, suggesting that observed resin acid alteration is mainly a biotic effect, in addition to known abiotic effects such as oxidation (Fig. S3)[23]. Further 2D-NMR analysis of the 2-week sample confirmed dehydroabietic acid as the main extractive, with agreement both of expected aromatic and carboxylic structural signals (Fig. S4). For the later week samples, 7-oxodehydroabietic acid was the main compound (by gas chromatography; GC) (Table S2). However, interfering signals in the sample and the relatively low concentration of 7-oxodehydroabietic acid confounded confirmation by NMR.

The majority of fatty acids in spruce bark are oleic and pinoleic acids, while nearly all sterols and steryl esters are free β-sitosterol and β-sitosterol linked to fatty acids[24]. Of these non-toxic compounds, the triglycerides were rapidly metabolized and almost entirely consumed after 2 weeks, while for fatty acids and steryl esters there was a sharp initial drop followed by a gradual decrease (Fig. 2). Such degradation has previously been shown for both fungi and bacteria[13,17]. Again, no abiotic effects were observed, such as spontaneous hydrolysis of steryl esters (Fig. S3). The sterols in the actively growing culture increased initially (weeks 0–8), and then instead decreased (weeks 8–24), which could both be derived from the microbial growth (membrane sterols) or release from steryl esters. The long-chain fatty acids and alcohols all decreased over time, though in the blank sample they increased,

possibly due to spontaneous oxidation (Fig. S3). Additionally, a ferulic ester compound could be identified in the later timepoints using different 2D-NMR methods, where the aromatic moiety as well as the ester bond were confirmed rather than a free ferulic acid (Fig. S5). Possibly, this ester is derived from partial degradation of suberin.

## Enrichment of resin acid degrading bacteria in spruce bark microbial communities

To compare the chemical changes in the bark to the development of the microbial consortium enriched on it, we analyzed its composition by long-read sequencing of marker gene regions of both bacteria (16S rRNA) and fungi (ITS2), followed by taxonomical assignment, and a principal component analysis showed good agreement between replicates (Fig. S6). After quality control and bioinformatic processing, an average of 64,163 ± 4612 fungal reads and 87,290 ± 4962 bacterial reads were generated per sample (Table S3), with mean sequence lengths of 495 ± 21 and 1359 ± 11 bp, respectively. The alpha diversity of the fungal community was lower than that of bacteria and overall did not change drastically during growth (Figs. 2, S7). However, the composition changed significantly over the course of the experiment, where Saccharomycetes drastically decreased in abundance compared to the original inoculum, whereas Eurotiomycetes and especially Tremellomycetes increased. Interestingly, both Agaricomycetes and Leotiomycetes were reduced in relative abundance in the first timepoints but began gradually increasing after 4 weeks and especially the proportion of Leotiomycetes increased, suggesting they are inhibited by extractives, or possibly by the species that metabolize these in early degradation stages. In the 2-week sample, the fungal community was dominated by *Apiotrichum* (31.3%), *Penicillium* (14%), *Trichoderma* (6.4%), and *Mucor* (6%), which are common cosmopolitan fungal genera.

The alpha diversity of the bacterial community was higher than that of the fungi throughout the experiment. In contrast to the fungal community however, it sharply decreased in the 2-week sample and then gradually increased to almost the initial value (Figs. 2, S7). This suggests an enrichment of key species that are likely involved in metabolism of the inhibitory bark substances, or species tolerant of these. In the inoculum, Pseudomonadota (Proteobacteria) dominated together with Actinomycetota (Actinobacteria) and a smaller fraction of Bacteroidota (Bacteroidetes), and with other phyla in lower abundance. During the colonization of fresh bark, the consortium was however completely overtaken by Pseudomonadota at 2 weeks, while Actinomycetota reappeared after 8 weeks together with Bacteroidota. While Pseudomonadota decreased somewhat in relative abundance after the initial colonization, it remained dominant throughout the experiment. Interestingly, the relative abundance of Bacteroidota gradually increased over the course of the experiment, which could indicate a shift toward carbohydrate utilization as this phylum is known to comprise carbohydrate degradation specialists[25]. Overall, the bacterial microbial community was dominated by the genera *Pseudomonas*, *Burkholderia-Caballeonia-Paraburkholderia*, and *Nitrospirillum* where the former two have previously been linked to resin acid degradation[18,19]. To gain further insight into the microbiome at 2 weeks, this sample was further shotgun metagenome sequenced.

## Metagenomics divulges potential key resin acid-degrading bacteria

The combined observation of resin acid degradation and lowered bacterial diversity during initial growth suggests enrichment of highly specialized microorganisms that can tolerate, detoxify, or fully metabolize resin acids. To gain deeper biological insight on the process, short-read metagenome sequencing was first performed on the 2-week sample, and later followed by long-read sequencing to improve quality and enable reconstruction of metagenome-assembled genomes (MAGs) (Table S4 and S5). Within the metagenome, a majority of reads

from the long-read sequencing were classified as bacterial (91%), with lower numbers for eukaryotes (3%), and the remaining reads classified as archaeal, viral, or unclassified reads (Fig. S8). 15 MAGs were binned resulting in 13 MAGs of high quality (>90% completion, <5% contamination) and two of medium quality with more than 5% contamination or no 16S rRNA (Table S5). As in the 16S rRNA analyses, the metagenome showed Pseudomonadota to be the most abundant phylum within the bacterial community and comprised of Alphaproteobacteria and Gammaproteobacteria (Fig. S8), representing either unknown species or genera including known resin acid degraders such as *Pseudomonas* and *Burkholderia*[19,22].

Resin acid-degradation is sparsely described in literature but within the genomes of *Pseudomonas abietaniphila* BKME-9[T], *Pseudomonas diterpeniphila* A19-6a, and *Paraburkholderia xenovorans* LB400[T] (formerly: *Burkholderia xenovorans*) a so-called *dit* gene cluster has been attributed to this function from gene knockout studies (Table S6)[19,26,27]. The presence of genes homologous to *dit* genes from *P. abietaniphila* and *P. xenovorans* in the MAGs was investigated (Figs. 3, S9), and a complete *dit* cluster was identified in MAG15 which was also the most abundant microorganism (19.2% based on reads) in the metagenome (Fig. 3). Varying *dit* cluster completeness was found in the other MAGs from BLAST comparisons (basic local alignment search tool; ≥30% seq. id.). Many of the MAGs classified as Pseudomonadota (MAG 1, 4, 6, 7, 8, 9, 14, 15) have a *dit*A1 homolog which is a gene shown to be essential for resin acid-degradation through gene knockout analyses in both *P. abietaniphila* and *P. xenovorans*[19,26]. Comparing all *dit*A1 homologs among the MAGs showed sequence identities around 30% for the vast majority, which corroborates previous studies suggesting that *dit*A1 is not predominantly shared via horizontal gene transfer[28]. In the metagenome, MAG7 was the second most abundant (12.6%) and classified as *Paraburkholderia*. It encodes a putative partial *dit* cluster comprising *dit*A1, *dit*B, *dit*D, *dit*L, and *dit*Q (Fig. S10). Despite MAG14 being the third most abundant and identified as *Pseudomonas*, the only putative co-located *dit* genes are a homolog of *dit*A1 and *dit*B interspersed with two genes of unknown function (Fig. S10). MAG6 was identified as *Paraburkholderia tropica* with an average nucleotide identity (ANI) score of 99.06%, which despite only being 4.3% relative abundant has a considerable amount of putative *dit* genes, including two *dit*A1 genes found in two different clusters, together with *dit*F and *dit*J homologs. Curiously, MAG4 encodes a seven-gene cluster comprising *dit*A1, *dit*A2, *dit*C (24.5% seq. id.), *dit*B, *dit*I, *dit*L, and *dit*Q, while only representing 2% of the metagenome reads.

While extractive degradation was the main activity of the community, we sought to identify potential carbohydrate-active enzymes (CAZymes) present in the MAGs using dbCAN2[29]. In total, 1783 putative CAZymes were identified, including 777 glycoside hydrolases (GHs), 752 glycosyltransferases (GTs), 135 carbohydrate esterases, 81 auxiliary activities, and 38 polysaccharide lyases (Fig. S11). Interestingly, the lowest abundant MAGs encoded the highest number of degradative CAZymes (non-GT), supporting our chemical analysis results that imply that carbohydrates are not the main nutrient during initial bark degradation (Figs. S11–S15). Overall, the abundance of predicted CAZyme activities was highest for microbial glycans and starch, and comparatively low for plant cell wall polymers except for pectin (Figs. S16, S17). MAG13 (0.7% relative abundance) stood out as a likely polysaccharide specialist from the Bacteroidota, and it encodes 115 predicted GHs, many of which are found in putative polysaccharide utilization loci (PULs)—large gene clusters comprising sugar capture-, transport-, degradation-, and regulatory proteins[25]. MAG13 encodes 37 of the PUL signature SusC/SusD-like pairs (sugar transport/capture), in ten different PULs (Fig. S18). Two of these PULs appear to target pectin (rhamnogalacturonan I side chains and rhamnogalacturonan II, respectively), two hemicelluloses (xyloglucan and galactoglucomannan, respectively), and one starch (Fig. S18), which are all

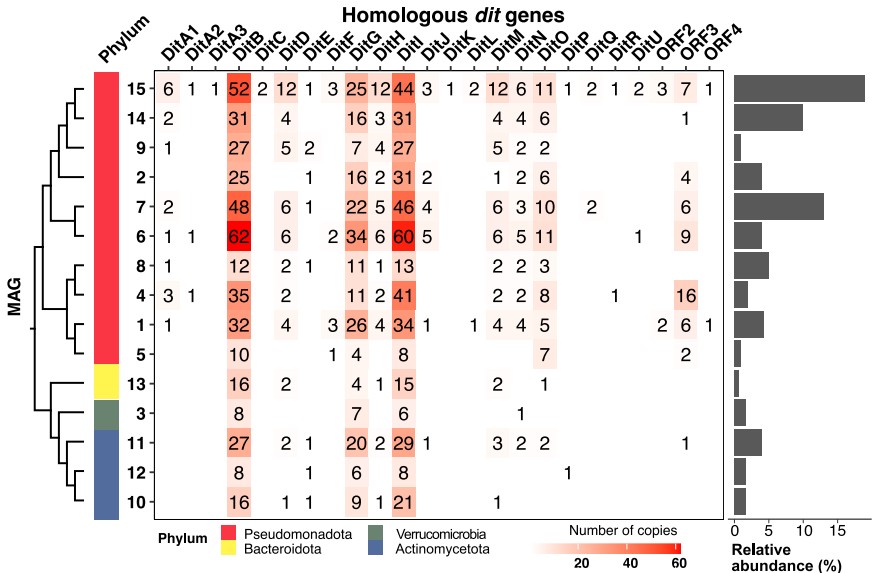

**Fig. 3 | Metagenome Assembled Genomes (MAGs) from the 2-week spruce bark degradation sample.** Proteins were identified in the MAG sequences by comparing with the *dit* clusters in *P. abietaniphila* BKME-9[T] and *P. xenovorans* LB400[T] using protein-protein BLAST with a *E* value of 1e[−10] and sequence identity ≥30%, and duplicate sequences with the same Dit protein hit and query were removed. Partial gene clusters (co-located genes) are found in Fig. S10. Taxonomic classification is based on GTDB (Genome Taxonomy Database), and phyla are indicated by color[71]. MAGs are clustered according to proteome-based GBDP distance (Genome Blast Distance Phylogeny) using the Type (Strain) Genome Server (TYGS)[32]. This type of phylogenetic clustering was used as no 16S rRNA sequence could be extracted from MAG12. The relative abundance of MAGs (determined via CoverM v. 0.6.1.) within the metagenome is shown on the right.

polysaccharides found in spruce bark[4,30,31]. The other PUL targets are not easily identified but could be extracellular polymeric substances from competing microorganisms.

## Isolation, genome sequencing, and phenotyping of the dominant resin acid-degrading bacterium

With resin acid degradation being a major activity in early spruce bark degradation and MAG15 being numerically dominant, we pursued to study this specific population in more detail. Based on the previous chemical and genomic information, we designed culture conditions with abietic acid as a carbon source which enabled us to successfully isolate a bacterial strain (designated PIA16) from the 2-week sample which was further characterized via genome-sequencing and phenotypic analysis. Long sequencing reads were assembled into a reference-grade, closed 6,715,763 bp genome, with a 62% G + C content and 5987 predicted protein-coding genes (Tables S7, S8). A significant Average Nucleotide Identity (ANI) match (>99%) between the isolate and MAG15 supported it being the same species (Fig. S19). Coverage estimations of the complete PIA16 genome in our resin-degrading enrichments (2-week sample) showed it to be 34.1% of the total metagenome, with CoverM default settings, and 1.24% with more stringent settings (99% identity), suggesting that the PIA16/MAG15 population likely represents a collection of highly similar strains exhibiting microdiversity but presumably sharing metabolic capabilities. PIA16 was found to be affiliated to the *Pseudomonas* genus, while taxonomic classification analyses using both the Genome Taxonomy[32] and SILVA[33] databases supported our strain being a new species, which we named *Pseudomonas abieticivorans* sp. nov. A phylogenetic analysis based on 16S rRNA similarity between *P. abieticivorans* and the most closely related *Pseudomonas* species, which do not include previously identified resin acid-metabolizing species, showed *P. abieticivorans* being located on a separate phylogenetic branch (Fig. S20), again supporting its classification as a new species. ANI analysis showed the most closely related species being *P. laurylsulfatiphila* DSM 105097[T] (81.9%), *P. baetica* a390[T] (81.4%), *P. putida* NBRC 14164[T] (81.6%), and *P. koreensis* Ps 9-14[T] (81.6%) (Table S9), well below the recommended 95% cut-off to distinguish species.

Differential phenotypic tests compared to closely related strains are shown in Table S9, and further differentiate *P. abieticivorans* biochemically from these. The ANI similarity between *P. abieticivorans* and species previously shown to degrade resin acids was also low, with values of 81.9%, 80.6%, and 80.15% to *P. vancouverensis*, *P. abietaniphila*, and *P. multiresinivorans*, respectively[18]. The final growth density of *P. abieticivorans* on resin acids was observed to be fairly low (maximum $OD_{600}$ ~ 0.2), and it is interesting to speculate what limits growth on these substrates, but as the solubility of resin acids is very low (1.7–5 mg/L), solubilization is likely a major factor limiting utilization[9].

Genome analysis of *P. abieticivorans* revealed few degradative CAZymes, and only suggests the ability to grow on starch, with enzymes from GH13 (7 homologs), GH31 (1), GH77 (1), in addition to several putative enzymes involved in bacterial cell wall lysis/remodeling (e.g., GH23 (5), GH73 (2), GH103 (2)) (Fig. S21). The API ZYM enzyme activity screen showed a negative result for β-galactosidase and β-glucosidase, but positive for esterase lipase (C8) and lipase (C14) (Table S9), which further supports the hypothesis that this species does not primarily consume polysaccharides but rather extractive compounds in spruce bark. Similar to other resin acid degrading isolates, *P. abieticivorans* can grow on both dehydroabietic and abietic acid as a sole carbon source (Fig. 4), but interestingly it can also grow on the more toxic isopimaric acid[9], which is an unusual trait (Fig. S22)[22]. To gain insight into correlations in protein-coding sequences between known resin acid degrading strains we performed a genome collinearity analysis (Table S10), identifying 15,765 (40.67%) sequences involved in 639 synteny blocks (Fig. S23), defined as a block of five orthologous genes found in the same order between the compared genomes[34]. As expected, fewer collinearity blocks were observed between *P. abieticivorans* and *Paraburkholderia xenovorans* compared to *Pseudomonas* strains. Interestingly, when analyzing the inter-species synteny blocks, it was observed that *P. abieticivorans* shares a high collinearity level in two regions (bases 5,144,228 to 5,171,342, and 6,497,666 to 6,541,998) which is composed of around 80 protein-coding genes (Fig. S23). When analyzing the number of times that a gene was included inside a synteny block and shared among species, it was observed that the region 0.7–4.7 Mb in the genome of

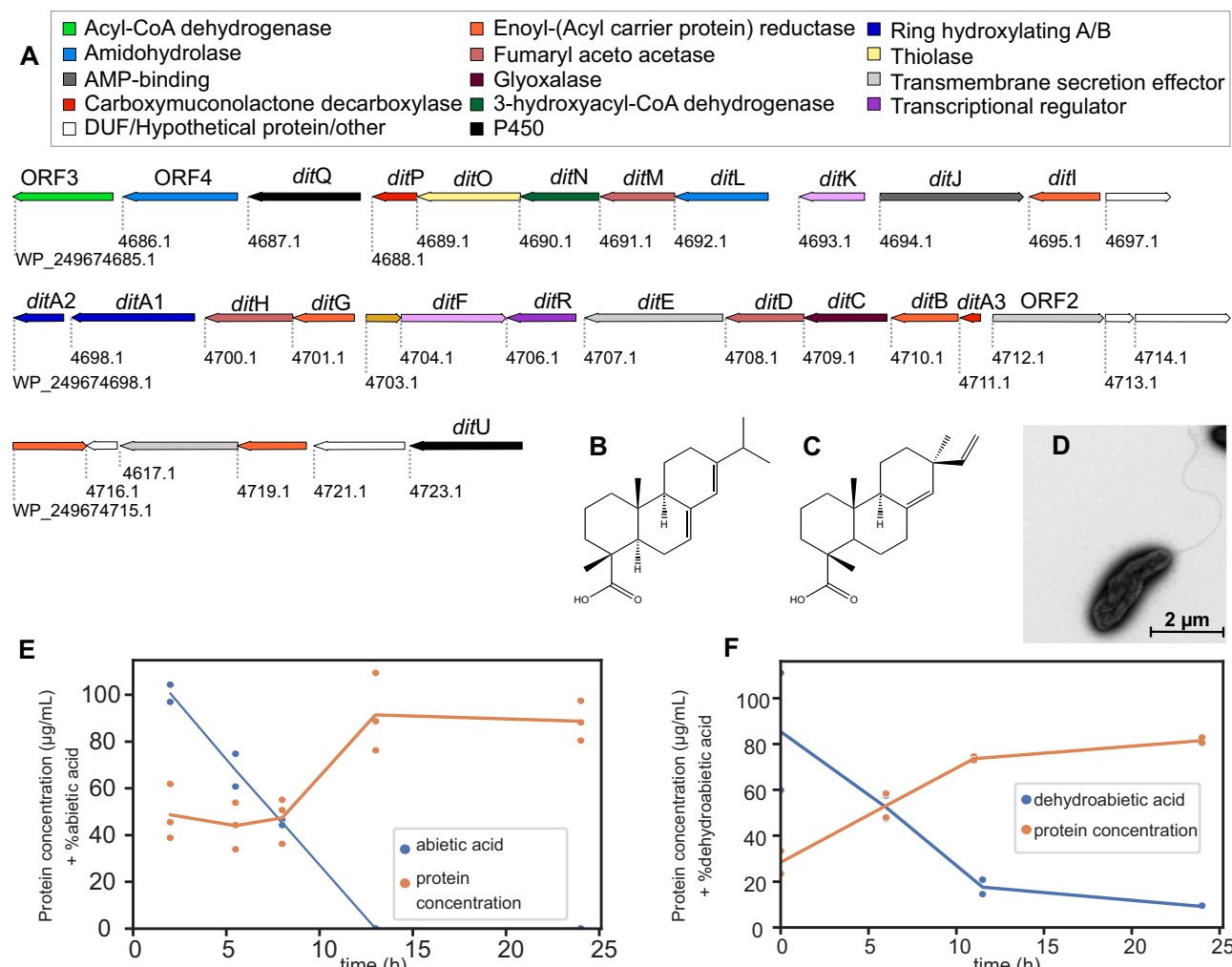

**Fig. 4 | The genetic organization of the *Pseudomonas abieticivorans* dit cluster and growth on resin acids. A** The identified *dit* cluster which spans 34.6 kbp. Genbank accession numbers are shown below each gene, gene names above, and their functional annotations according to pfam are color coded (Table S11; DUF−domain of unknown function, AMP−adenosine monophosphate). The resin acids are divided into two subtypes: abietane and pimarane, the former represented by (**B**) abietic acid, and the latter by (**C**) isopimaric acid. **D** Representative transmission electron microscopy image of a *P. abieticivorans* cell, showing the presence of a single flagellum (from 30 images including several cells each). Degradation of resin acids by *P. abieticivorans* on (**E**) abietic acid and (**F**) dehydroabietic acid, and concomitant cellular protein concentrations as a proxy for growth as optical density monitoring is complicated by the insoluble nature of the resin acids. Data points are based on duplicate or triplicate biological experiments and lines show the means.

*P. abieticivorans* is frequently shared among the *Pseudomonas* species studied here. One such shared synteny block contained clusters of genes previously identified to sustain growth on resin acids (Fig. S23).

As expected from the metagenomics, *P. abieticivorans* encodes a complete *dit* cluster (24/24 expected genes), which was compared to those found in known resin acid-degrading strains using BLAST and collinearity analyses (Fig. 4, Table S11). Strikingly, *P. abieticivorans* had an identical *dit* cluster genetic organization as that of *P. abietaniphila*, though only partial overlap was observed with *P. xenovorans* (Fig. S23). A lower number of co-located *dit* genes (10/24) was also identified in *P. resinovorans*, but in *P. multiresinovorans*, homologs to *dit* genes were not found in a cluster but rather spread out in the genome.

## Discussion

Bark is vital for the protection of trees and disruption of this outer barrier by e.g., wood-boring insects can lead to enhanced microbial access, decimated coniferous forests worldwide, and disruption of global ecosystems[35,36]. Despite its fundamental protective role, how bark can be degraded by microorganisms is not known. Previous studies have shown that microbial communities differ between bark and

wood, as well as the specific sampling point on the tree, and also that the presence of bark slows down the degradation of the underlying wood[37]. The importance of the bacterial communities that exist on bark of living trees was recently highlighted, where it was shown that bark-dwelling methanotrophic bacteria decrease methane emissions from trees, thus mitigating global warming[38]. Fungal bark communities have been less investigated but shown to change over time as wood degrades through measurements of wood decay stage, density, dry matter content, and C:N ratio[12]. Despite the importance of extractives in the antimicrobial properties of bark, they have, however, previously been overlooked.

In this study, we monitored the development and dynamics of a bark-sourced microbial community on fresh spruce bark over 6 months, as well as the chemical changes to the material to identify important degradation stages, compounds, and microorganisms for this process. The minor degradation of polysaccharides was surprising, as especially non-cellulosic and non-crystalline carbohydrates are regarded as readily degradable by plant biomass-metabolizing enzymes, while slow lignin degradation was observed as expected. It is worth pointing out that some of the measured monosaccharides

likely originate from the microorganisms, as these could not be separated from the bark itself, and possibly the acid insoluble residue could partially comprise microorganism-derived molecules in addition to lignin. It appears that for the microbial enrichment used in this study, the bark extractives are the main nutrients in the initial degradation stages, or possibly inhibit the growth of specialized carbohydrate-degrading species. The initial sharp decrease in alpha-diversity in the bacterial community, and that it only recovered after the resin acids had begun being metabolized, suggests that resin acids in particular may act as "gate-keeper" molecules and that their detoxification is a key step in bark degradation. This agrees well with resin acids being regarded as major defensive compounds with broad antimicrobial properties[8].

Resin acids are thought to mainly be encapsulated within resin canals in wood[39], and the apparent increase in the concentration of especially dehydroabietic acid at the initial degradation stage is interesting. It could indicate that the resin acids are in fact inaccessible in fresh bark and possibly weakly bound to other molecules, polymerized, or otherwise tightly encapsulated within the bark before being released. Currently, there is no information on resin acids being for instance ester-bonded to any structural component within the bark, but it could be speculated that their release is instead linked to the degradation of triglycerides, as these can help solubilize the hydrophobic resin acids in the resin itself. Recently, extractive compounds have been shown to either inhibit or enhance the activity of carbohydrate-active enzymes, such as lytic polysaccharide monooxygenases[40], so it is possible that the extractives may exert general enzyme-inhibitory rather than toxic effects that block wider carbohydrate utilization.

Over a longer time frame, the amplicon data suggest that the bacterial populations begin shifting from an extractive focus to a carbohydrate conversion, given the increase of Bacteroidota[25], though a clear shift in actual carbohydrate-degrading capacity in the enrichment did not become apparent during our study. Direct comparison with previous studies is not trivial as they typically have analyzed single time points, combined bark and wood, and/or omitted detailed chemical analyses[10–12]. However, in some studies, we see similar profiles of microbial abundance as reported here, with a dominance of Pseudomonadota for bacteria[11] and mainly basidiomycetes and ascomycetes for fungi[11,12], though without chemical analyses of the bark itself knowing what stage of degradation these previous studies correspond to is not possible. Without complementary techniques, such as quantitative metaproteomics, the biological roles of the fungi remain more elusive as the number of reads from sequencing does not necessarily reflect biological importance, and fungi are often regarded as main drivers of boreal forest biomass conversion. Follow-up studies of bark-degrading microbiomes will be crucial to determine how diverse such communities can be, and whether our results reflect general trends for microbial spruce bark metabolism. While the studied cultures were fully exposed to air, we cannot exclude the possibility that (micro) anaerobic pockets were also formed in the material during the experiment, though as full conversion of resin acids has not been observed anaerobically[22], such micro-environments would likely not be highly abundant. The dry matter content used in the experiment (30%) was close to that of fresh spruce bark (typically around 35–40%)[41,42], which is not known as an anaerobic tissue.

Our deeper metagenomic analysis revealed the abundance of specialized MAGs and their *dit* cluster-related genes, assumed to encode enzymes and proteins involved in resin acid (diterpene) degradation[19,22,26], which further indicates that resin acid metabolism is crucial to thrive on fresh spruce bark. Interestingly, the number of encoded degradative CAZymes in our reconstructed MAGs was relatively low in the MAGs of highest abundance (Figs. 3, S11), which might explain the observed low carbohydrate turnover. Our metagenomic analysis also facilitated the discovery and subsequent characterization

of *P. abieticivorans*, which dominated our enrichment cultures and encoded a complete *dit* cluster while appearing to have a limited capacity for complex carbohydrate metabolism. The observation that other MAGs did not encode full *dit* clusters might indicate that these organisms have incomplete resin acid degradation pathways, unknown pathways, or are in fact unable to fully metabolize resin acids. The molecular basis for resin acid degradation has not been demonstrated apart from one study of a cytochrome P450 monooxygenase from *Streptomyces griseolus*[43], though whether resin acids are a natural substrate for this enzyme is unclear. Further investigation of *P. abieticivorans* and related species can help shed light on this important process, which likely involves solubilization of the resin acids at some stage to facilitate import into the cell.

Overall, our study indicates that microbial degradation of the bark is a slow process, and also that abiotic changes to the material are minor even over our 6-month experiment. Spruce is indispensable to the forest industry and is used for pulp and paper production in the northern hemisphere. Of the millions of cubic meters of spruce harvested annually, the bark comprises a significant fraction which today is poorly utilized; its low dry matter content and ash content make burning inefficient but its polymers and especially the high proportion of extractive compounds with varying properties could potentially be utilized to develop new products. Understanding how bark can be degraded by different microorganisms is key for such future valorization, for instance using select enzymes acting on valuable extractives, and can also be of high relevance to limit the harmful effects of wood-boring insects. Our isolated new *Pseudomonas* species, or newly discovered strains thereof, may for instance become a useful source of especially resin acid-modifying enzymes to selectively introduce new functional groups on these interesting and generally toxic molecules. Bark from other species contains other types of extractives, with for example birch bark being enriched in betulin and pine in condensed tannins[5], and our study sets a basis for investigating whether degradation also of other bark types progresses via initial degradation of key "gatekeeper" molecules that strongly inhibit the growth of but a few extractive-detoxifying microbial species.

## Methods

### Bark preparation

Spruce bark was obtained from the Iggesund pulp and paper mill (Iggesund, Holmen AB, Sweden), from a bark pile resulting from stripping of spruce logs at the mill after harvest, with the average age of trees at harvest being ~70 years. The bark was left to dry at room temperature for seven days. Thereafter, the bark was milled in a Wiley-type mill, mesh size <0.1 mm, and was afterwards sterilized with 25 kGy by gamma-irradiation (Mediscan GmbH & Co KG). To validate sterilization, bark samples were subjected to 0, 10, 20, and 30 kGy, respectively and then added to either liquid lysogeny broth (LB) or potato dextrose agar plates. No growth was observed from bark samples that had received 20 or 30 kGy after several weeks of incubation.

### Microbial cultivation

To partially mimic growth conditions on spruce bark in nature, solid-state cultivation was used. First, a bark inoculum was generated by growing unsterilized, milled bark in 2.3 mL M9 minimal medium (disodium hydrogen phosphate 6.778 g/L, potassium phosphate 5 g/L, ammonium chloride 1 g/L, sodium chloride 0.5 g/L, magnesium sulfate 0.24 g/L, calcium dichloride 0.012 g/L) per g bark for 6 months (16 h light, 8 h dark per 24 h, ~21 °C), to obtain an actively growing culture (evaluated from visual growth). 1 g (wet weight) of the ~30 g growing culture was added to 10 g sterile bark samples with 2.3 mL M9 medium per g bark, yielding a dry matter content of ~30%, somewhat below bark sampled on trees (~35–40%)[41,42]. The solid-state cultures were incubated aerobically in the same conditions as the inoculum, above,

and individual samples were collected after 0, 2, 4, 8, 12, 16, 20, and 24 weeks of growth in biological triplicate experiments. The moisture was adjusted if needed using water based on regular weighing of the samples. At sampling, ~4 g bark samples (dry weight) were weighed and used for chemical analysis of the carbohydrates, lignin, and extractives present in the bark, while the remaining culture was frozen at −20 °C and used for 16S rRNA and ITS profiling. Non-inoculated blank samples sterilized by gamma-irradiation in the same manner as the inoculum treated sample were used as controls.

### Soxhlet extraction
For evaluation of the effect of microbial growth on the spruce bark extractives, Soxhlet extraction was used. The bark samples were extracted with 200 mL acetone as solvent, in order to extract the majority of extractive compounds, with 10 min per extraction for 24 cycles. After extraction, the acetone was evaporated using rotary evaporation until 50 mL solvent remained, then left in a fume hood for evaporation of remaining solvent. The percentage of acetone-soluble extractives was determined gravimetrically based on the weight of the starting material.

### Ash content
For evaluation of the effect of microbial growth on the ash content present in the bark, the ash concentration was measured according to National Renewable Energy Laboratory standard method[44]. In brief, the weight of the inorganic residue left after 1 g of bark sample (after drying at 105 °C overnight) had been subjected to dry oxidation at 575 °C was measured.

### Chemical characterization of carbohydrates and lignin
To elucidate if the microorganisms were degrading carbohydrates or lignin in the spruce bark, the acetone-extracted bark was dried overnight, followed by hydrolysis using 72% sulfuric acid according to established protocols[45]. The insoluble residue was determined gravimetrically. The filtrate from the hydrolysis was used to determine monosaccharide and acid-soluble lignin content. The acid-soluble lignin was determined by absorbance measurements at 205 nm on a Specord 205 (Analytic Jena), using an extinction coefficient of $110 \, dm^3 g^{-1} cm^{-1}$[46]. The monosaccharide composition of the bark was monitored using high-performance anion-exchange chromatography with pulsed amperometric detection (HPAEC-PAD) on an IC5000 system (Thermo Scientific). A 2 × 250 mm Dionex Carbopac PA1 column (Thermo Scientific) with a 2 × 50 mm guard column (Thermo Scientific) was used at a column temperature of 30 °C. The eluents were A – water, B – 200 mM NaOH, and C – 200 mM NaOH and 170 mM sodium acetate[47]. The samples were eluted using eluent A (0.26 mL/min) and detected using a post-column addition of eluent B (0.13 mL/min). Thereafter, the column was washed with eluent C and equilibrated with eluent A. The monosaccharide concentrations were determined using an internal fucose standard and pure external standards of arabinose, mannose, galactose, glucose, and xylose. Peak analysis was performed using Chromeleon software 7.2.10 (Thermo Scientific).

### Extractive group analysis using GC-FID and GC-MS
To determine the effect of microbial growth on the lipophilic extractive groups present in the spruce bark acetone extracts, the bark extractives were analyzed by GC coupled to a flame ionization detector (FID) and GC-mass spectrometry (MS) by MoRe Research Örnsköldsvik AB. The bark extracts were derivatized and analyzed as trimethylsilyl derivatives. The internal standards (heneicosanoic acid, cholesterol heptadecanoate, and 1,3-dipalmitoyl-2-oleyl glycerol), at a concentration of 200 μg/mL, were added to 50 μL samples containing 30 mg/mL of bark extractives dissolved in acetone. Thereafter, the vials were evaporated under nitrogen and derivatized using 100 μL N,O-Bis(trimethylsilyl)trifluoroacetamide (BSTFA), 50 μL trimethylchlorosilane (TMCS), 20 μL pyridine, and heated for 20 min at 70 °C.

The concentrations of extractive groups (fatty acids, resin acids, sterols, steryl esters, and triglycerides) were evaluated on a Thermo Trace 1300 chromatography system coupled to an FID using an on-column Programmable Temperature Vaporization (PTV) injection method. The temperature program of the oven started at 100 °C, which was held for 1.5 min, followed by increasing the temperature to 340 °C using a gradient of 10 °C/min, and holding this temperature for 3 min. The PTV temperature increased from 80 °C to 340 °C at 10 °C/min, which was held for 10 min. The fatty acids, resin acid, and sterol ester concentrations were evaluated on a HP-5ms (30 m, 0.25 mm internal diameter, 0.25 μm film thickness) column relative to the peak area of the internal standard heptadecanoic acid. High boiling compounds, such as steryl ester and triglyceride concentrations were evaluated using a HP-1 SIM/Dist (5 m, 0.53 mm internal diameter, 0.15 μm film thickness) column based on the relative peak area towards the internal standards cholesteryl heptadecanoate and 1,3-dipalmitoyl-2-oleylglycerol, respectively. Peak evaluation was performed using Chromeleon software (Thermo Fisher Scientific).

GC-MS was used for the identification of individual compounds in the fatty acids, resin acids, and sterols present in the bark extracts. Quantification of the individual compounds was based on the FID total ion peak, relative to the peak area of the internal standard heptadecanoic acid. GC-MS analysis was performed on an Agilent 8890 chromatography system with a CTC Analytics CombiPAL (Autosampler) coupled to an Agilent 5975 C MSD mass spectrometer with split-less injection using the same column and as the long column FID experiment, a HP-5ms (30 m, 0.25 mm internal diameter, 0.25 μm film thickness) column and helium as a carrier gas at a rate of 1.2 mL/min. The temperature program of the oven started at 100 °C, was held for 1 min, thereafter, the temperature was increased to 340 °C at 10 °C/min, which at the end was held for 3 min. The injector was kept at 230 °C. Compounds were identified in the MS by comparing their mass spectra with those of the Wiley and NIST libraries on MassHunter (Agilent).

### Extractive analysis using NMR
All NMR measurements were conducted on a Varian Inova 500 MHz operating at 11.7 T with a 5 mm HFX-probe. Measurements were performed at 298 K with acetone-d6 as solvent. The $^1$H-measurements were conducted using an 8 μs $^1$H-detection pulse, 2 s acquisition time, 5 s recycle delay, and 32 scans. $^{13}$C-measurements were conducted using a 14 μs $^{13}$C-detection pulse, 1 s acquisition time, 5 s recycle delay, and 8096 scans. Correlation spectroscopy was recorded using 2048 scans with 256 increments. Heteronuclear single quantum coherence was recorded using 64 scans with 384 increments and a transfer delay of 3.425 ms corresponding to a 145 Hz J-coupling. Heteronuclear multiple bond coherence was recorded using 64 scans with 512 increments and a transfer delay of 62.5 ms, corresponding to an 8 Hz J-coupling. For all samples, the solvent was used as chemical shift reference with the acetone-d6 methyl signal referenced to 2.05 ppm and 29.84 ppm for $^1$H and $^{13}$C, respectively.

### DNA extraction for sequencing of fungal and bacterial gene regions
For each timepoint, duplicate spruce bark samples were subjected to DNA extraction and community analysis performed by DNASense ApS (Denmark) using the standard protocol for FastDNA Spin kit for Soil (MP Biomedicals) with the following exceptions: 500 μL of sample, 480 μL sodium phosphate buffer, and 120 μL MT buffer were added to a Lysing Matrix E tube, and bead beating was performed at 6 m/s for 4 × 40s[48]. Gel electrophoresis using Tapestation 2200 and Genomic DNA screentapes (Agilent) was used to validate product size and purity

of a subset of DNA extracts, with the majority of the DNA being >5000 bp and of good quality for amplicon sequencing. DNA concentration and purity were measured using Qubit dsDNA HS/BR Assay kit (Thermo Fisher Scientific).

## Library preparation and sequencing of bacterial and fungal marker gene regions

Amplicon libraries for the bacteria 16S rRNA gene variable regions 18 (bV18A) and fungi ITS2 were prepared using a custom protocol, and performed by DNASense ApS. Up to 25 ng of extracted DNA was used as template for PCR amplification, and each PCR reaction (50 μL) contained 0.2 mM dNTP mix, 0.01 units of Platinum SuperFi DNA Polymerase (Thermo Fisher Scientific), and 500 nM of each forward and reverse primer in the supplied SuperFI Buffer. PCR was done with the following program: Initial denaturation at 98 °C for 3 min, 25 cycles of amplification (98 °C for 30 s, 62 °C for 20 s, 72 °C for 2 min) and final elongation at 72 °C for 5 min. The forward and reverse primers used include custom 24 nt barcode sequences followed by the sequences targeting bV18A: [8F] AGRGTTYGATYMTGGCTCAG and [1391R] GACGGGCGGTGWGTRCA[48–52]. The forward and reverse primers used include custom 24 nt barcode sequences followed by the sequences targeting the fungi ITS2: [ITS29] TACACACCGCCCGTCG and [ITS4] TCCTSCGCTTATTGATATGC[53].

The resulting amplicon libraries were purified using the standard protocol for CleanNGS SPRI beads (CleanNA, NL) with a bead-to-sample ratio of 3:5. DNA was eluted in 25 μL of nuclease-free water (Qiagen, Germany). Sequencing libraries were prepared from the purified amplicon libraries using the SQKLSK114 kit (Oxford Nanopore Technologies, UK) according to the manufacturer's protocol with the following modifications: 500 ng total DNA was used as input, and CleanNGS SPRI beads for library cleanup steps. DNA concentration was measured using Qubit dsDNA HS Assay kit (Thermo Fisher Scientific, USA). Gel electrophoresis using Tapestation 2200 and D1000/High sensitivity D1000 screentapes (Agilent, USA) was used to validate product size and purity of a subset of amplicon libraries.

The resulting sequencing library was loaded onto a MinION R10.4.1 flowcell and sequenced using the MinKNOW 23.04.6 software (Oxford Nanopore Technologies, UK). Reads were basecalled and demultiplexed with MinKNOW guppy g6.5.7 using the super accurate basecalling algorithm (config r10.4.1_400bps_sup.cfg) and custom barcodes.

## Bioinformatic processing bacterial gene regions

The sequencing reads in the demultiplexed and basecalled fastq files were filtered for length (320–2000 bp) and quality (phred score >15) using a local implementation of filtlong v0.2.1 with the settings –min_length 320 –max_length 2000 –min_mean_q 97. The SILVA 16S/18S rRNA 138 SSURef NR99 full-length database in RESCRIPt format was downloaded from the QIIME on 29 September 2022[33,54,55]. Potential generic placeholders and dead-end taxonomic entries were cleared from the taxonomy flat file, i.e., entries containing uncultured, metagenome or unassigned, were replaced with a blank entry. The filtered reads were mapped to the SILVA 138.1 99% NR database with Minimap2 v2.24r1122 using the ax Mapont command[56] and downstream processing using samtools v1.14[57]. Mapping results were filtered such that query sequence length relative to alignment length deviated <5%. Noteworthy, low-abundant operational taxonomic units (OTUs) making up <0.01% of the total mapped reads within each sample were disregarded as a data denoising step. Further bioinformatic processing was done via RStudio IDE (2023.3.0.386) running R version 4.3.1 (20230616) and using the R packages: ampvis2 (2.8)[48], tidyverse (2.0.0), seqinr (4.2.30), ShortRead (1.58.0), and iNEXT (3.0.0)[58,59].

## Bioinformatic processing fungal gene regions

The sequencing reads in the demultiplexed and basecalled fastq files were filtered for length (320–2000 bp) and quality (phred score >15)

using a local implementation of filtlong v0.2.1 with the settings –min_length 320 – max_length 2000 –min_mean_q 97. The filtered reads were mapped to the QIIME formatted and 99% identity clustered UNITE database release 9.0[60] with Minimap2 v2.24r1122 using the ax mapont command[56] and downstream processing using samtools v1.14[57]. Potential generic place holders and dead-end taxonomic entries were cleared from the taxonomy flat file, i.e., entries containing uncultured, metagenome or unassigned, were replaced with a blank entry. Mapping results were filtered such that alignment length was >125 bp and mapping quality >0.75. Noteworthy, low abundant OTUs making up <0.01% of the total mapped reads within each sample were disregarded as a data denoising step. Further bioinformatic processing was done as for the bacterial gene regions, above.

## Metagenome-assembled genomes—DNA extraction and sequencing

DNA from the week 2 sample was extracted using the DNeasy Power-Soil Pro Kit following the manufacturer's recommendations (Qiagen, Germany), by DNASense ApS. For Oxford Nanopore sequencing, a custom SPRi-bead size-selection protocol was further used to clean up and size-select (approx. cut-off 1500–2000 bp) DNA prior to sequencing. Gel electrophoresis on Genomic DNA ScreenTapes and using the Tapestation 2200 (Agilent, USA) was used to validate DNA size distribution of a subset of DNA extracts. DNA concentration and purity were measured with the Qubit dsDNA HS Assay kit (Thermo Fisher Scientific, USA) and on the NanoDrop One (Thermo Fisher Scientific, USA).

For Illumina sequencing, the DNA was quantified using Qubit (Thermo Fisher Scientific, USA) and fragmented to ~550 bp using a Covaris M220 with microTUBE AFA Fiber screw tubes and the settings: duty factor 10%, peak/displayed power 75 W, cycles/burst 200, duration 40 s and temperature 20 °C. The fragmented DNA was used for metagenome preparation using the NEB Next Ultra II DNA library preparation kit. The DNA library was paired-end sequenced (2 ×151 bp) on a NovaSeq S4 system (Illumina, USA). For Oxford Nanopore sequencing, a long-read sequencing library was prepared according to the SQK-LSK110 protocol (ONT, Oxford, United Kingdom). Approximately 50–100 fmole was loaded onto primed FLO-MIN106D (R9.4.1) flow cells and sequenced on the GridION platform using MinKNOW Release 22.05.7 (MinKNOW Core 5.1.0). Raw Illumina reads were filtered for PhiX using Usearch11[61] and trimmed for adapters using Cutadapt v. 3.7[62]. Rasusa v. 0.6.1 was used to subset the Illumina data to 100 gb (gigabases)[63]. Raw Oxford Nanopore Fast5 files were basecalled in Guppy v. 6.1.5 using the Super-accurate basecalling algorithm. Basecalled fastq were was then adapter-trimmed in Porechop v. 0.2.4 Porechop using default settings. NanoStat v.1.6.0[64] was used to assess the quality parameters of the basecalled data. The trimmed data were then filtered in Filtlong v. 0.2.0 with min_length set to 1000 bp and min_mean_q set to 95.

A short-read metagenome was assembled with Megahit v. 1.2.9 using a predefined kmer-list (43,71,99,127)[65]. 16S rRNA gene sequences were extracted using Barrnap v. 0.9 and classified against the SILVA SSU 138.1 database[33]. Forward reads were classified with Kaiju v. 1.8.2 against the nr_euk 2021-02-24 database[66] and visualized with Krona v. 2.8.1[67].

Draft metagenomes were assembled with Flye v.2.9[68] by setting the metagenome parameter (– meta). The assembled metagenome was subsequently polished with quality-filtered Oxford Nanopore data using one round of polishing with Minimap2 v. 2.24[56] and Racon v.1.4.20[69] and two rounds of polishing with Medaka (v.1.6.1). The metagenome was finally polished with Minimap2 v. 2.24[56] and Racon v.1.4.20[69] using Illumina data. The metagenome was visualized in RStudio IDE (4.2.0 (2022-04-22)) running R version 4.0.4 (2021-02-15) and using the mmgenome2 R package v. 2.1.3. Metagenome-assembled genomes (MAGs) were manually extracted in mmgenome2 v. 2.2.1.

Completeness and contamination values were assessed using CheckM v. 1.1.3[70]. MAG abundance was calculated using CoverM v. 0.6.1. MAGs were classified using the Genome Taxonomy Database toolkit v. 2.1.0 against the r207_v2 Genome Taxonomy Database[71,72]. Ribosomal RNA genes were identified and extracted using Barrnap v. 0.9. MAGs were annotated using Prokka v. 1.14.6[73].

## Analysis of metagenome-assembled genomes

Putative resin acid-degrading proteins present in the MAGs were identified by using Basic Local Alignment Search Tool (BLAST) with protein sequences from *dit* cluster genes from *Pseudomonas abietaniphila* BKME-9 and *Paraburkholderia xenovorans* (Table S6) against the MAGs using an E-value of 1e$^{-10}$ and sequence identity ≥30%. Putative carbohydrate active enzymes were identified using dbCAN2 HMMdb v10[29]. Functional annotation of protein family databases (Pfam) was performed in eggNOG-mapper using usegalaxy.eu[74,75].

## Isolation and identification of resin acid-degrading bacteria

Resin acid degrading bacteria were isolated by serial dilution from one sample after 2 weeks of bark degradation, that had been stored at −20 °C, using plates containing *Pseudomonas* Isolation Agar (PIA) (20.0 g/L beef extract peptone, 1.4 g/L magnesium chloride, 10 g/L potassium sulfate, 13.6 g/L agar, 20 mL/L glycerol). 5 g of the partially degraded bark was diluted in 45 mL of water, left to soak for 30 min at room temperature, and 100 mL inoculum from the resulting liquid was spread onto the plates with tenfold dilutions up to 10$^{-5}$. Plates were incubated at room temperature for 5 days. Obtained colonies were re-streaked on M9 minimal medium agar plates (15 g/L agar) containing 10 g/L of abietic acid. Single colonies were obtained by sequential streaking on fresh abietic acid plates, and for initial identification these were picked, boiled for 10 min, and used for PCR amplification of 16S rRNA and the signature *Pseudomonas* gene *rpoD*. Each PCR reaction (50 mL) contained Phusion polymerase (Thermo Scientific), 5X HF buffer (Thermo Scientific), water, 1 mL of boiled colonies, and 10 nM of each forward and reverse-tailed primer T primers 27F (5′-AGAGTTT-GATCMTGGCTCAG-3′) and 1470 R (5′-TACGGYTACCTTGTTACGACT T-3′), and the partial *rpoD* sequence using primers PsEG30F (5′-ATY-GAAATCGCCAARCG-3′) and PsEG790R (5′-CGGTTGATKTCCTTGA-3′), as previously described[76]. PCR was conducted with the following program: initial denaturation at 94 °C for 20 s, 35 cycles of amplification (94 °C for 20 s, 55 °C for 30 s, 72 °C for 60 s) and a final elongation at 72 °C for 7 min. The PCR products were analyzed by gel electrophoresis, purified, and sequenced using Sanger sequencing (Macrogen). Sequence results were identified using BLAST at NCBI[77]. An isolate, designated PIA16, was selected for genome sequencing and physiological tests, based on its ability to grow on abietic acid and sequence dissimilarity (16S rRNA and *rpoD*) to known species. The species was later named *Pseudomonas abieticivorans*, which is used henceforth. *P. abieticivorans* sp. nov. is deposited at the Culture Collection University of Gothenburg (CCUG 76343T) and Deutsche Sammlung von Mikroorganismen und Zellkulturen GmbH (DSM 114633).

## Genome sequencing of *Pseudomonas abieticivorans*

DNA extraction, sequencing, and analysis were performed by DNASense ApS. DNA was extracted using the DNeasy PowerSoil Pro kit following the manufacturer's recommendations (Qiagen) with minor modifications. Gel electrophoresis using Tapestation 2200 (Genomic DNA and D1000 screentapes, Agilent) was used to validate product size and purity of a subset of DNA extracts. DNA concentration was measured using the Qubit dsDNA HS Assay kit (Thermo Fisher Scientific). A rapid (SQK-RBK 110.96) library was prepared according to the manufacturer's protocol, except for minor modifications (Oxford Nanopore Technologies). The library was loaded on a primed R9.4.1 MinION flow cell and sequenced on the GridION platform using

MinKNOW Release 21.10.4. Sample DNA concentrations were measured using the Qubit dsDNA HS kit and the DNA quality was evaluated using TapeStation with the Genomic ScreenTape (Agilent Technologies). Sequencing libraries were prepared using the NEB Next Ultra II DNA library prep kit for Illumina (New England Biolabs) following the manufacturer's protocol. Library concentrations were measured in triplicate using the Qubit dsDNA HS kit and library size estimated using TapeStation with D1000 HS Screen-Tape. The sequencing libraries were pooled in equimolar concentrations and diluted to 4 nM. The samples were paired end sequenced (2 × 301 bp) on a MiSeq (Illumina) using a MiSeq Reagent kit v3, 600 cycles (Illumina) following the standard guidelines for preparing and loading samples on the MiSeq. Fast5 raw data were basecalled and demultiplexed in Guppy (Oxford Nanopore Technologies) v. 6.0.1 using the sup algorithm. Basic read statistics including amount of data produced, median read quality and read N50 were assessed with NanoPlot v. 1.36.2[64]. Raw Illumina reads were filtered for PhiX using Usearch11[61] and trimmed for adapters using cutadapt (v. 2.8[62]). A draft de novo assembly was produced with Flye v. 2.9-b1768[68]. The draft assembly was subsequently polished one time with Racon v. 1.4.20[69] and Minimap2 v. 2.22-r1101[56] and twice with Medaka v. 1.4.3 (Oxford Nanopore Technologies). The genome was finally polished with Minimap2 v. 2.22-r1101[56] and Racon v.1.4.13[69] using Illumina data. Assembly graph and draft assembly coverage were visualized and inspected with Bandage v. 0.8.1[78]. Genome annotation was conducted using Prokka (https://github.com/tseemann/prokka) v. 1.14.6[73]. Sample genomes were classified with the GTDB (Genome Taxonomy Database) tool Kit v. 1.7.0[71] against the GTDB r202[72]. The genome was functionally annotated (COG) using WebMGA[79].

## Phylogenetic and collinearity analysis

To compare the similarity between *P. abieticivorans* and the most closely related known species, a phylogenetic analysis of all the *Pseudomonas* species reported in NCBI: GenBank[80] was performed. BLASTn on NCBI web server[80] was used to acquire 16S ribosomal RNA sequences, with *Pseudomonas* (taxid:286) sequences database selected, and limited to a maximum of 1000 target sequences. Multiple sequence alignment was executed using CLUSTALW with *msa* package[81] in R v4.2.2[82]. Alignments were processed by using bios2msd[83] and *adegenet*[84] R packages. Phylogenetic trees were constructed using *ggtree*[85] and *ggplot2*[86] R packages. Tree estimation was calculated by the neighbor-joining method supported by ape[87] package. Collinear analysis of *P. abieticivorans* compared to five species reported to be able to utilize resin acids (Table S10) was performed in MCScanX[88]. Protein-coding sequences were used. A local BLAST database was built for each species, to further run BLASTp[89] with E-value less than 1e$^{-10}$ and maximum five hits. In all the cases, *P. abieticivorans* was set as query. Default parameters were kept running MCScanX. Input files and output file analysis were carried out in R. Circos v0.69-9[90] and used to visualize the collinear results.

## Morphology, physiological, and biochemical tests

Negative staining and transmission electron microscopy (TEM) were used to examine the morphology of *P. abieticivorans*. Formvar-coated copper grids (150 mesh) were rendered hydrophilic by glow discharge before applying 5 μL of cell suspension (OD$_{600}$ = 0.5 in lysogeny broth; LB) for 3 min. The suspension was removed by blotting onto Whatman filter paper and the remaining bacteria were immediately fixed by applying 5 μL of 2% glutaraldehyde for 2 min. After blotting away the fixative and washing the grids twice with distilled water, the samples were stained for 1 min with 1% uranyl acetate. Excess stain was removed by blotting and the grids were air-dried before imaging. The samples were examined with a Talos L120C transmission electron microscope (Thermo Fisher Scientific) operating at 120 kV, and images were captured with a 4k × 4k BM-Ceta CMOS camera.

Growth at various temperatures (4, 20, 30, 37, 45 °C) was investigated on LB plates over 4 days and growth was assessed based on the presence of visible colonies. Growth based on salt concentration (NaCl 0–12% w/v) and the pH-growth range was assessed in LB medium by monitoring $OD_{600}$ after 4 days of growth, with pH adjusted using NaOH or HCl. Fluorescence was studied by plating on King A or King B media (Millipore). Additional physiological tests were conducted using the API 20 NE and API ZYM systems according to the manufacturer's instructions (bioMérieux). All tests were performed in duplicate.

## Chemotaxonomic analysis

Analysis of cellular fatty acids was carried out by DSMZ Services, Leibniz-Institut DSMZ – Deutsche Sammlung von Mikroorganismen und Zellkulturen GmbH (Braunschweig, Germany). The fatty acid methyl esters of the PIA16 strain were obtained from cells grown in tryptic soy broth (Millipore) at 28 °C for 24 h. Cellular fatty acids were analyzed after conversion into fatty acid methyl esters by saponification, methylation, and extraction using minor modifications of established methods[91,92]. The fatty acid methyl esters mixtures were analyzed by GC-FID using Sherlock Microbial Identification System (MIS) (MIDI, Microbial ID). Peaks were automatically integrated, and fatty acid names and percentages were calculated by the MIS Standard Software (Microbial ID).

## Growth on resin acids and resin acid analysis

Growth on dehydroabietic acid and abietic acid was done in cultures of M9 medium containing 100 mg/L of dehydroabietic acid (Megazyme ~90%) or abietic acid (Sigma ~75%) and inoculated with a starting optical density at 600 nm ($OD_{600}$) of 0.01. The concentration of the resin acids (100 mg/L) was deliberately kept low to minimize background as resin acids are present as insoluble particles in the medium. Prior to inoculation, *P. abieticivorans* was grown in 10 mL LB medium at room temperature overnight. To follow growth and resin acid degradation over time, 1 mL of culture was sampled for evaluation of growth using protein concentration and degradation of resin acids using GC-MS. Intracellular protein concentration was used as a pseudo-value for growth due to the resin acids' low solubility in water, making OD measurements unreliable. Briefly, cells were spun down, the pellet was resuspended in 1 M NaOH, boiled, and neutralized using HCl. Protein concentration was measured using the Bio-Rad BCA kit following the manufacturer's instruction. Resin acid extracts were analyzed using a modified method according to ref. 93. Briefly, the resin acids were acidified using 40 μL HCl and spiked with an internal standard of 0.1 mg isopimaric acid (Carbosynth) and extracted three times using 1 mL ethyl acetate. Resin acids were dried over nitrogen and resuspended in 500 μL acetone and silylated using 25 μL BSTFA and TMCS (Sigma). Resin acid identification and quantification were performed by capillary GC (Agilent 7890A). Helium, the carrier gas, had a flow rate of 1 mL/min and the MS source was operated at 230 °C with the quadrupole at 150 °C. Analytes were separated using a HP-5 column with a temperature program starting at 70 °C, which was held for 2.25 min, was increased to 200 °C at 20 °C/min, thereafter 5 °C/min until the temperature reached 230 °C. The final ramp was at 35 °C min to 300 °C which was held for 10 min. Injector temperatures were at 300 °C. Helium, the carrier gas, had a flow rate of 1 mL/min and the MS source was operated at 230 °C with the quadrupole at 150 °C. The NIST MS Search Programme (Vers. 2.2) was used for identification using the library NIST/EPA/NIH Mass Spectral Library (NIST 11). Semi-quantification of the resin acids was done as using the internal standard and the following relationship: $WS = AS \times WI/AI$, where $W$ stands for the mass fraction, $A$ for the area in the chromatogram, I the internal standard (isopimaric acid) and S the species, respectively.

The growth of *P. abieticivorans* on different types of resin acids was additionally analyzed using a Growth Profiler (Enzyscreen), using continuous imaging and pixel-based growth determination, in a 24-well plate (6 × 4) using 100 mg/L of each resin acid: dehydroabietic acid (Megazyme ~90%), abietic acid (Sigma ~75%), and isopimaric acid (Carbosynth). Growth was screened at 30 °C, and the cultures inoculated from an overnight culture of *P. abieticivorans* grown on LB broth to a starting $OD_{600}$ of 0.01 in a total volume of 1.2 mL.

## Protologue−description of *Pseudomonas abieticivorans* sp. nov

*Pseudomonas abieticivorans* (a.bi.e.ti.ci.vo′rans. N.L. neut. n. acidum abieticum, abietic acid; L. pres. part. vorans, eating; N.L. part. adj. abieticivorans, eating abietic acid). Cells are Gram-negative, rod-shaped (0.77 μm wide, 2.6 μm long), with a singular flagellum, and catalase and oxidase negative. Growth occurs at 4-30 °C. Optimal growth temperature is between 25 and 28 °C. The species grows at pH 5-9 and in the presence of 6 w/v NaCl. The species is negative for reduction of nitrates to nitrate and reduction of nitrates to nitrogen, for indole production, fermentation of glucose, urease, and proteases for gelative hydrolysis. The species is positive for assimilation of D-glucose, L-arabinose, D-mannitol, potassium gluconate, capric acid, malate, and trisodium citrate, and negative for assimilation of D-mannose, *N*-acetyl-D-glucosamine, D-maltose, adipic acid, and phenylacetic acid. The enzymatic activities which were positive were alkaline phosphatase, esterase (C4), esterase lipase (C8), leucine arylamidase, valine arylamidase, naphthol-AS-BI-phosphohydrolase, acid-phosphatase, arginine dihydrolase, and catalase. The ones which displayed a negative test were lipase (C14), cystine arylamidase, trypsin, α-chymotrypsin, β-galactosidase, β-glucuronidase, α-glucosidase, β-glucosidase, *N*-acetyl-glucosaminidase, α-mannosidase, α-fucosidase. Growth is detected in R2A, PIA, and LB. The major fatty acids are summed feature 3 ($C_{16:1}$ ω7c and/or $C_{16:1}$ ω6c, 26.12%), summed feature 8 ($C_{18:1}$ ω6c and/or $C_{18:1}$ ω7c, 13.96%), $C_{16:0}$ (31.57%), $C_{17:0}$ cyclo (12.79%), $C_{12:0}$ (4.11%), $C_{12:0}$ 2-OH (3.05%), $C_{12:0}$ 3-OH (3.49%), $C_{10:0}$ 3-OH (3.00%), $C_{14:0}$ (0.47%). The type strain, PIA16 (=CCUG 76343T, =DSM 114633T) was isolated from spruce bark on abietic acid. The G + C content of the type strain is 61.1 mol% and the genome size is 6,715,763 bp.

## Reporting summary

Further information on research design is available in the Nature Portfolio Reporting Summary linked to this article.

## Data availability

All sequencing reads have been deposited at the National Center for Biotechnology Information (NCBI) under BioProject ID PRJNA912085 and PRJNA803013. The 16S and ITS reads have been deposited in the Sequence Read Archive (SRA) under the accession numbers SRX21853827-SRX21853858 (Table S12). PIA16 reads have been deposited under the accession numbers SRR24684300 and SRR24684301 (Biosample accession SAMN25609858), and the metagenome reads under accession numbers SRR22729505 and SRR22729506 (Biosample accession SAMN32218794). The recovered MAGs have been deposited in GenBank under the accession numbers SAMN32241433-SAMN32241447 (Table S13). The *P. abieticivorans* PIA16 genome has been deposited in GenBank under the accession number GCA_023509015.1, the 16S rRNA sequence under the accession number ON945571.1, and the *rpoD* sequence under the accession number OP594298. Other data generated or analyzed during this study are included in the Supplementary Information files, and Source data are provided with this paper.

## Code availability

All the datasets and code used in this study for phylogenetics 16S rRNA and collinearity analysis are available at DOI:10.5281/zenodo.7596797.

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

## Acknowledgements
Funding was provided by an energy-oriented basic research grant from the Swedish Energy Agency and the Swedish Research Council (project number 46559-1) and a research project grant from the Carl Trygger Foundation (CTS 21:1424), awarded to J.L., as well as two grants from the Adlerbertska Foundations, awarded to A.S.R. and J.L. The Novo Nordisk Foundation (grant no. NNF20CC0035580 and project no. 0054575) is also acknowledged for funding to E.J.K and P.B.P, respectively.

We would like to thank Dr. Marcel Taillefer from Chalmers University of Technology, Henrik Kjeldal and Rasmus Wollenberg from DNASense for helpful discussions, and Holmen AB for providing spruce bark. We acknowledge the Centre for Cellular Imaging at the University of Gothenburg and the National Microscopy Infrastructure, NMI (VR-RFI 2019-00022) for assistance with electron microscopy.

## Author contributions
Conceptualization, A.S.R., J.L., P.B.P.; Formal analysis, A.S.R., S.V., A.I., M.H., A.T.R.; Funding acquisition, A.S.R., M.H., J.L.; Methodology, A.S.R., M.H., P.B.P., J.L.; Project administration, A.S.R., J.L.; Resources, A.S.R., M.H., J.L.; Supervision, J.L., M.H.; Validation, A.S.R., A.T.R., M.H., J.L.; Visualization, A.S.R., A.I., A.T.R., S.V., J.L.; Writing—original draft, ASR; Writing—review & editing, A.S.R., A.I., A.T.R., E.J.K., M.H., J.L., P.B.P., S.V.

## Funding

## Competing interests
The authors declare no competing interests.
