## [Peer Review File · Nature Communications]

Resin acids play key roles in shaping microbial communities during degradation of spruce barkReviewer #1 (Remarks to the Author):

This study focuses on spruce bark degradation as a consequential but understudied microbial decay process in nature and of value to industry. The work uses a mesocosm-type approach with clever enrichment-like culturing to create treatments and time points over a decay sequence to culture, track bark chemical changes targeting extractives (high % in bark, relative to wood), and track community dynamics in fungi (ITS 2 primers) and bacteria/archaea (v4 16S) using metagenomics. A *Pseudomonas abieticivorans* likely involved in resin acid degradation is also reported, newly-described, with deeper evaluation of that new isolate.

I think this work is valuable, and the 'omics' portions are deep and very cool, but there are a couple of problems I had with drawing conclusions with the set-up, most importantly that the mesocosm approach has forced an environment that does not represent the most common, aerobic bark environment in nature. The moisture content at 2.3 mL per g of solid would 'swamp' wood by 3x, and bark should be no different - I fear this has created an anaerobic system, that your Agaricales 'bloom' in later decay stages attests to this, and that you are going deep into one aspect 'resin degradation' that may be a pretty small fraction of the overall story. I think this paper is actually better focusing on the new *Pseudomonas* isolate deep work, perhaps finding a better way to frame the mesocosms. Otherwise, you have a lot of deep omics work resting on a mesocosm design that I think is reflecting something like the bottom of a pond rather than the forest floor.

Specifics:

Intro

58 – The specific example of spruce here seems orphaned by some kind of bigger picture number. I think you have a purpose for mentioning the 10-15% of harvest volume....but it is bark...so if it is used in products (I assume not) it would not matter much about its degradation. If it is left in the forest, than your study becomes more consequential. If 10-15% of each tree is left on the forest floor while the 85-90% is taken to the sawmill, that is 100% bark left behind...then, your bark story is everything, at least in a spruce clearcutting operation. I'd frame it that way.

61- Fig 1 seems unnecessary. It is not super helpful for delineating spatial information for readers, and the chemical components are just pictures instead of words, not specific to anything. I'd remove it.

67 – remove extractives = improved saccharification for any lignocellulosic, true?

89 – remove 'curiously only'

94 – links to carbon cycling and valorization would be stronger if you fix the introduction P1 as per line 58 comment

Methods

429 – Is 2.3 mL per g anaerobic conditions? Bark degradation in nature will likely be a very aerobic process, just like wood. At 0.8/1 water to wood, wood is too wet for most fungi. This is 2.3/1 which is almost 3x higher. It will be anaerobic, I assume based on the description.

438 – sterilized?

441 – earlier mention of hydrophobic and hydrophilic extractives, but only acetone used here...does that miss some compounds?

455 – how was 205 nm determined? Acid-soluble lignin from different sources determines different wavelengths selected. I would add citation here.

463 – hydration factors to recapitulate polymer concentrations?

465 – I see no compensation to the 'illusion of gain' issue when mass is removed from solid substrates. If no lignin were removed, but 100% of carbohydrates were removed, the lignin concentration would go up despite no lignin being formed. Must multiply by fraction of original.

569 – DNA extraction from a lignocellulosic with this high % extractives and then using for metagenome assembly. Did quality warrant doing this? Qubit and Nanodrop, but results/QC?

Results

Fig 2 – similar to fig 1, I just don't think this is helping us much. Maybe in supplementals.

124 – Ash will be a very small percentage here, it looks from S1 like 5% (high relative to wood ash), so 53% gain means 20 mg/g or so gain. Also the S1 shows a big difference between the time zero for 'blank' vs. 'sample' that, if averaged, would make the 53% more like 21% I think.

S1 – While I am inspecting S1, I would note that a similar major disparity between time zero materials but with error bars showing significant differences is in the lignin, with non-inoculated

being 30% and inoculated 'biotic' treatment being 40%. That, to me, would indicate problems in one of two things: 1) your acid hydrolysis technique, or 2) replication or actual, physical sample size issues leading to noise that, for some reason, is not reflected in error bars. 30% lignin vs. 40% lignin....that is very large disparity in material from same source at time zero stage.

137 – if you had presented the bark in its natural state, you could have simply measured mass loss and had your answer. Instead, you used ash content to normalize data to try to deal with the 'illusion of gain', and I'd be uncertain about conclusion here.

177 – I think these specifics are interesting, for sure. Looking within a fraction at loss patterns.

191 – Or they are inhibited in anaerobic conditions, not extractives, but as your samples dried, were able to outcompete others.

359 – Or your conditions were conducive to anaerobic, resin-degrading bacteria.

401 – The *Pseudomonas abietivorans* discovery is cool. I think you've got an interesting bacteria in hand to work with, I just find this part to be something different from the rest of the study. It struggles to link.

Reviewer #2 (Remarks to the Author):

Dear Authors,

thank you for opportunity to review your manuscript about degradation of resin acids. I agree that the topic of microbial bark degradation is overlooked while being relatively important for forest ecosystem and for handling of waste from wood industry. The authors performed characterization of microbial community which was found to be growing on decomposing bark over the course of 6 months, extensively described chemical changes of the substrate and isolated bacterium which is able to grow on resin acids.

Generally, I like the extensive character of the study and the level of data reporting. Manuscript contains well prepared figures, a lot of supplementary data which might be suitable for interested reader. Reproducibility seems feasible as most of the raw data are given and Zenodo archive contains full folder/file structure which was used for analysis. Long read sequencing surely improved overall quality of metagenomic and genomic datasets and made the re-analysis of data more attractive for scientific community.

I certainly understand the temptation to present all the obtained data. While this is fine in most cases, I see also part which is irrelevant in the context of the current manuscript. I feel that fungal data which are mentioned only once in discussion (diversity, 359) and did not make it into abstract might be omitted without losing any key information from the manuscript.

Despite the aim to present all the data, a few important parameters of the study are not clear. These are replication design and clustering parameters of amplicon datasets (see line by line comments). Introduction part and Methods section are well-written. Although Results and Discussion parts are separated, several parts in Results would better fit in Discussion. Probably from this reason, the discussion is rather brief.

Overall, the presented work shows novelty in terms of researched topic while utilizing standard, widely applied techniques. The level of progress in the areas of forest ecology/carbon cycle/microbial role is moderate. Together with my opinion that the manuscript is not ready for publication in Nature Communications in its current form, I recommend its rejection.

Please see line by line comments and note that not all of them are negative.

59 - Bark mass estimate is worth of mentioning also here, not only in discussion.

77, 608, 616 - There are more resin degrading genera, but *Pseudomonas* was targeted in downstream workflow by medium and primers for marker gene. Could you please elaborate on that?

106-112 - not fitting to Results

126, Fig S2 - I do not see reason for ash standardization. At least monosaccharides are partly burnt to CO₂ which makes standardization useless. Per g of dry mass standardization is enough.

138 - Chitinolytic bacteria feeding on fungi might also represent some "microbial degradation". This is rather side note, the sentence is fine, although a little bit bold.

184, Table S3 - Table contains ASVs, although ASV generation is not mentioned in the Methods.

185 - 194 - Are fungi indeed needed in the present manuscript?

216 - Calculation of mean and SD based on duplicates is problematic from my point of view.

224 - 229, Fig S8 - Was annotation based on metagenomic contigs or reads? Description of such annotation is missing in Methods. Annotation of eukaryotic metagenomic contigs is very problematic due to their fragmentation.

232-235, 240-243, 257-261, 298-300, 305-309 - not fitting to Results, I think that any discussion should be avoided here since Discussion and Results are separated.

237 - this is nice finding!

285 - MAG15 with 69 contigs are used as proxy for abundance of identical isolate PIA16. It would be interesting to assess the abundance of PIA16 itself by mapping metagenomic reads to genome. It is complete bacterial genome, one contig, the best representation of the target organism.

296, 665 and Table S11 - Are there reference papers for other five resin acid degrading bacteria? I recommend to include them.

377 - Inverse relationship can be tested, right? I suggest to perform such test.

400 - After successful publication, please consider to send description of type strain to the staff of IJSEM in order to publish the new name also effectively.

417 - Please consider to use superscript "T" for type strain throughout the manuscript.

423 - Is it possible to note the age of source trees for bark? Is it important at all?

429 - Just to clarify - unsterilized AND milled bark was used, right? What were the temperature and light conditions during inoculum growth?

434 - What were the light conditions?

435 - This is one of my main concerns. Please clarify "biological triplicate experiments". I imagine that biological replicates represent bark from more than one experimental site or bark of differing age or differing culturing conditions or another grouping. If the bark came from one batch and cultivations were identical then all cultivations are technical replicates. This is relevant also for line 512 - DNA extractions where duplicates are not mentioned. SRA records show duplicated datasets for each sample and for each marker gene. Fig 3 and Fig 5 captions mention biological duplicates. Fig S1 and Fig S2 mention biological duplicates and two technical replicates. I found the replication description confusing, could you please clarify it?

554 - Could you please specify if any ITS extraction method was used?

566 - Could you please specify metagenome sampling timepoint here? It appears in Fig 2 caption and on line 208.

578 - Probably, this is not the case but based on current sentence it seems that fragmented DNA was used also for long read sequencing. Please clarify.

582 - Out of curiosity, could you please explain reason for subsampling raw metagenomic data?

604 - Which version of dbCAN was used? I suggest to report it.

738 - minor issues with data deposition, deposition itself is good - in order to be consistent with NCBI accessions, I recommend to use "Run" accession for SRX18691017 and SRX18691016 (SRR22729505 and SRR22729506, respectively). SRR22745721-SRR22745728 are not described in this paragraph although they are present in BioProject, these are probably ITS data. SRR23100117 is metagenomic dataset without further description, could you please add it? Could you please deposit also raw genomic data of *P. abieticivorans* PIA16?

Fig 5 - shows detailed analysis of gene content, I like this representation of data, it shows the depth to which authors analyzed data.

Reviewer #3 (Remarks to the Author):

In manuscript, entitled "Resin acids play key roles in shaping microbial communities during degradation of spruce bark", the authors claim to have analysed key stages in degradation of bark, an abundant renewable resource, and how bark defensive extractive compounds have major roles in shaping microbiomes.

While contributing with important information about the evolution of certain compounds of the bark along the development of certain inoculated microbial strains, which were previously artificially cultivated, the work misses the opportunity to analyze how bark is truly degraded in nature by natural microbial communities and how this natural communities truly evolve.

I have two major concerns:

1. To set up the experiment, the authors produce a culture of microbes which proceed from bark cultivated in M9 medium for 6 months. I wonder why they do this. When bark is moved from its original environment and it is cultivated under laboratory conditions, at certain specific temperature (not fluctuating), with no day/night variations, the microbial communities are going to completely change compared to the natural ones. Moreover, it seems by the figure 2 (even if not included in the methods) that the authors use agar for this cultivation, which can be used as solely nutrient provision for many microbes, so it even modifies more the microbial communities originally present in the bark.

During the culturing period, and at those different conditions, some of the taxa are going to overgrow, and some of them can produce and excrete different molecules that can inhibit other microorganisms or, by the contrary, to promote them. Also, some serendipitous microbes, which could even be as spores over the bark and maybe not developing in natural conditions, can grow in those artificial ones.

Therefore, the inoculated microbial community is going to be completely different to the originally present in nature, and that will impact not only the analysis of microbial communities, but also the degradation of bark compounds, compared to what it happens in nature. In other words, the way that the experiment was set implies that no matters if the original source to obtain the inoculum is bark or any other source, because the inoculated community is going to be completely different to the natural one.

For these reasons, I consider that the authors did not choose the best approach for their analyses. Why didn't the authors just evaluate the naturally present communities and analyze them and the extractives over the time?

2. Regarding the description of a new species, the closest related type strain to PIA16 according

the sequence of the 16S rRNA gene is *Pseudomonas laurylsulfatiphila* DSM 105097 (16S rRNA gene: NIRS01000004). Both strains share 99.18% similarity in this gene. As far as I can see (the phylogenetic tree is huge and has a very small letter size), this species has not been considered for the phylogeny -and it makes not sense to include such a big number of unrelated species in the tree-. Anyways, it has not been estimated the ANI value between PIA16 and *Pseudomonas laurylsulfatiphila* DSM 105097. The calculation of the ANI value with this strain is required in order to decide if PIA16 is a new species, since the similarity in the 16S with this strain is much higher than that with the type strains considered in the MS, and also it should be included in the phylogenetic analysis.

Other comments:

Line 58: "Spruce is one of the dominant species....in the northern hemisphere" TREE species? northern hemisphere FORESTS?

Line 63: Reference 6, Peng & Roberts test toxicity of bark extractives in Metazoa, I believe that it is not a good reference for microbial inhibition. Instead, there are other good articles referring antimicrobial potential of bark extractives.

Line 67-68: "Moreover, removal of extractive compounds from softwood bark has been shown to enhance its saccharification". I do not see the point of this sentence/information in here.

Lines 68-70: Some important works related to bark degradation by microbes (not only fungi, but also bacteria) have been omitted here. The consideration of these studies is important in order to really see to which extend your work add knowledge.

I can refer a few of them, but there are also several others:

-Malik, R.J., Trexler, R.V., Eissenstat, D.M. et al. Bark decomposition in white oak soil outperforms eastern hemlock soil, while bark type leads to consistent changes in soil microbial composition. *Biogeochemistry* 150, 329–343 (2020). <https://doi.org/10.1007/s10533-020-00701-7>.

-Hagge Jonas, Bässler Claus, Gruppe Axel, Hoppe Björn, Kellner Harald, Kraus Franz-Sebastian, Müller Jörg, Seibold Sebastian, Stengel Elisa and Thorn Simon 2019Bark coverage shifts assembly processes of microbial decomposer communities in dead wood *Proc. R. Soc. B*. 2862019174420191744 <http://doi.org/10.1098/rspb.2019.1744>

-Dong Wang, Olatunji Olusanya Abiodun, Jinlan Xiao, Wenqiang Zhao, Contrasting responses of microbial diversity and community structure in decaying root bark and xylem to N addition in an alpine shrubland, *Soil Biology and Biochemistry*, Volume 178, 2023, 108937, ISSN 0038-0717, <https://doi.org/10.1016/j.soilbio.2022.108937>.

-Weedon, J. T., Cornwell, W. K., Cornelissen, J. H. C., Zanne, A. E., Wirth, C., & Coomes, D. A. (2009). Global meta-analysis of wood decomposition rates: A role for trait variation among tree species? *Ecology Letters*, 12, 45–56. <https://doi.org/10.1111/j.1461-0248.2008.01259.x>

Line 131: "However, a larger effect could be seen for glucose (Fig. S1 & Fig. S2)" Please, slightly comment this effect (do not only refer to the figures). Also related to this: How do you explain the increase in glucose content within the uninoculated control over the time? (Figures S1 and S2). Once you make your study considering an initially sterilized bark, have you tested if the bark was really sterile? Is the gamma radiation killing microbes inside the bark? This should be tested. A possible presence of microbes in the negative controls could be the reason of the glucose increase over the time.

Lines 133-134 and lines 144-145 are discussion, not results.

Line 235: The presence of some dit genes within distant MAGs could indicate some event of horizontal transfer to fulfil the adaptation to the bark environment. However, the presence of incomplete (no-functional) dit clusters would not be so advantageous and could be just a transient gene gain. Have you compared the sequence of the dit genes among MAGs to see whether they are divergent or not? Have you searched for transposable elements within these contigs? Is the relative abundance of the dit genes concordant with the relative abundance of each MAG? These issues could help to gain a better perspective of the dynamics of these genes within the microbial community.

This paper exemplify a similar issue based on another gene cluster:
<https://www.nature.com/articles/s41467-018-04955-6>

Line 257: CAZyme categories are too broad to make correct assumptions. I suggest to also include an activity-based categorization and see whether some CAZyme groups related with the use of certain bark-polyssacharides are more abundant than others. I suggest to use the Figure S1 of the following paper as guidance for this classification:
<https://journals.asm.org/doi/full/10.1128/msystems.00829-22>

Line 345: add DEGRADATION after bark

Line 418: =DSM 114633 is missing a T (of type strain) in the end of the code.

Line 511: "DNA extraction for fungal and bacterial gene regions" as it is, this title does not make sense.

Lines 562-563: SILVA database, release 132 is a bit old version of the database. Currently the 138 version is being used.

Line 597: ribosomal and not ribosomale

Line 658: "known species" is super broad! You probably mean known *Pseudomonas* species. In any case, please check validated species in bacterio.net. Also, as I mentioned before, *Pseudomonas* is such an enormous genus that the inclusion of distant species does not make sense in order to evaluate the possibility of *P.abiericivorans* being a new species. You can leave the broad analysis if you want to show the position regarding other "resin acid" (as you name them) *Pseudomonas* species, but for the species description a more reduced tree including just related species (and all of the closest ones) is required.

Line 604: which version of dbCAN?

Description of new species within the genus *Pseudomonas* usually includes the analysis of respiratory quinones.

Fig S3 is missing the description of the F panel.

Response to reviewers

We would like to thank the reviewers for their constructive comments and we realized that parts of our experimental setup and the knowledge gap that we have addressed were not fully clear. We have revised the text to more explicitly describe that the biological/biochemical degradation process of spruce bark is not known today as previous studies have not tracked this process over time in detail. Without such fundamental knowledge as a basis, it would be very difficult to take samples of communities and bark composition and make detailed interpretations, at least in terms of degradation stages. In our work, we have instead followed the process in detail throughout six months and mapped both the chemical changes imparted by our microbial enrichment as well as the species diversity at each sampling point to paint a more holistic picture. We believe that this work will lay a foundation for future studies investigating spruce bark degradation, but also as a starting point to map the metabolism of barks from other species. We also believe that there is a great interest in society/industry in how such more unconventional renewable biomass types can be utilized.

We have addressed each comment, point-by-point, and provide answers below each in blue text. We provide both a new clean version of the manuscript, and a marked-up version. In addition to changes based on the reviewers' suggestions, we have also corrected a few typos that were found. Line numbers are indicated where appropriate, and refer to the new version of the manuscript. We believe that our manuscript has been much improved both in terms of clarity and quality, and we hope it is seen as worthy of publication in Nature Communications.

On behalf of all the authors,

Johan Larsbrink

Reviewer #1 (Remarks to the Author):

This study focuses on spruce bark degradation as a consequential but understudied microbial decay process in nature and of value to industry. The work uses a mesocosm-type approach with clever enrichment-like culturing to create treatments and time points over a decay sequence to culture, track bark chemical changes targeting extractives (high % in bark, relative to wood), and track community dynamics in fungi (ITS 2 primers) and bacteria/archaea (v4 16S) using metagenomics. A *Pseudomonas abieticivorans* likely involved in resin acid degradation is also reported, newly-described, with deeper evaluation of that new isolate.

I think this work is valuable, and the 'omics' portions are deep and very cool, but there are a couple of problems I had with drawing conclusions with the set-up, most importantly that the mesocosm approach has forced an environment that does not represent the most common, aerobic bark environment in nature. The moisture content at 2.3 mL per g of solid would 'swamp' wood by 3x, and bark should be no different - I fear this has created an anaerobic system, that your Agaricales 'bloom' in later decay stages attests to this, and that you are going deep into one aspect 'resin degradation' that may be a pretty small fraction of the overall story. I think this paper is actually better focusing on the

new *Pseudomonas* isolate deep work, perhaps finding a better way to frame the mesocosms. Otherwise, you have a lot of deep omics work resting on a mesocosm design that I think is reflecting something like the bottom of a pond rather than the forest floor.

Response: This is a relevant comment, and certainly if we had used solid wood, we would have a very wet material. The bark structure is however much more porous than wood, and the volume we chose to add to it was based on reaching ~70% moisture. On the tree, the bark may have a moisture of around 60-65% (e.g. <https://doi.org/10.1051/epjconf/20123302005> or <https://doi.org/10.1016/j.indcrop.2013.10.009>), but when fallen onto the forest floor or gathered in a pile as in industrial settings this is bound to be higher, so we aimed to increase the moisture somewhat to ensure good growth. Below we show an image of the starting material, after addition of medium, to illustrate that the material is far from swamped. The images shown were taken after the addition of media as described in the methods (the actual experiment was performed in covered flasks to avoid contamination). As can be seen, the bark mass was fully exposed to air. There still is a possibility that (micro)anaerobic pockets were formed in our solid state cultivations, and we agree that it is important to mention this and have added text around this in the discussion. To clarify both the moisture content and the possibility of partial anaerobicity, we have updated the manuscript as follows:

Discussion, lines 391-394: “While the cultures were fully exposed to air, we cannot exclude the possibility that (micro)anaerobic pockets were also formed in the material during the experiment, though as full conversion of resin acids has not been observed anaerobically²², such micro-environments would likely be minor.”

Methods, lines 462-464: “1 g (wet weight) of the approximately 30 g growing culture was added to 10 g sterile bark samples with 2.3 mL M9 medium per g bark, yielding a moisture content of ~70%, is slightly above bark sampled on trees (~60-65%⁴⁵).”

Specifics:

Intro

58 – The specific example of spruce here seems orphaned by some kind of bigger picture number. I think you have a purpose for mentioning the 10-15% of harvest volume....but it is bark...so if it is used in products (I assume not) it would not matter much about its degradation. If it is left in the forest, than your study becomes more consequential. If 10-15% of each tree is left on the forest floor while the 85-90% is taken to the sawmill, that is 100% bark left behind...then, your bark story is everything, at least in a spruce clearcutting operation. I'd frame it that way.

Response: thank you for this comment to help us clarify this part of the text, and it was also partly raised by Reviewer 3. The bark represents 10-15% of the total volume at harvest, but as you say the main target in forestry is the wood and the huge associated volumes of bark are largely seen as waste. It is important to understand what happens to the bark both from a fundamental angle, but also an industrial as this large renewable resource could likely be used better. We have rephrased the text to add this industrial perspective, and hope we now address both the industrial and the biological perspective, as understanding how bark can be degraded is important in both cases, either for future valorization or as a fundamental process.

Revised text:

Lines 58-62: “Spruce is one of the dominant tree species in northern hemisphere forests and of high economic value, and its bark comprises around 10-15% of the volume at harvest. In the Nordic countries alone, bark is produced in close to 400 million m³ per year⁵, but it is largely seen as an industrial side stream and is either left to rot in the forest or burnt for energy at the mill.”

Lines 414-418: “Of the millions of cubic meters spruce harvested annually, the bark comprises a significant fraction which today is poorly utilized; its high moisture and ash content make burning inefficient but its polymers and especially the high proportion of extractive compounds with varying properties could potentially be utilized to develop new products.”

61- Fig 1 seems unnecessary. It is not super helpful for delineating spatial information for readers, and the chemical components are just pictures instead of words, not specific to anything. I'd remove it.

Response: we believe a general overview of the differences between wood and bark would be very useful for the reader as this is a somewhat “unconventional” biomass material to study and the differences would be good to point out also visually. We have updated the figure and caption to better highlight the differences between wood and bark (mainly extractive- and ash content) and hope it is more positively received.

Figure 1. Overview of the structure of tree bark compared to wood. The bark can be classified as inner or outer, which depends on its proximity to the external environment and is governed to a large extent by its age. Compared to regular wood, the bark is enriched in minerals (ash) as well as the class of molecules referred to as extractives, which have highly variable properties. In the box, representative lipophilic extractive groups in spruce bark are shown: A) triglycerides (glycerol linked to the three unsaturated fatty acids octadecenoic acid (C18:1); octadecadienoic acid (C18:2); octadecatrienoic acid (C18:3), B) resin acids (dehydroabietic acid), and C) sterols (R=H, β -sitosterol) and steryl esters (R=alkyl chain, e.g. octadecanoic acid).

67 – remove extractives = improved saccharification for any lignocellulosic, true?

Response: in line also with a comment from Reviewer 3, the sentence has been removed.

89 – remove ‘curiously only’

Response: done.

94 – links to carbon cycling and valorization would be stronger if you fix the introduction P1 as per line 58 comment

Response: yes we agree, please see response above.

Methods

429 – Is 2.3 mL per g anaerobic conditions? Bark degradation in nature will likely be a very aerobic process, just like wood. At 0.8/1 water to wood, wood is too wet for most fungi. This is 2.3/1 which is almost 3x higher. It will be anaerobic, I assume based on the description.

Response: as mentioned in our response to the first general comment, above, we do not judge the material to be anaerobic.

438 – sterilized?

Response: yes, it was treated as the other samples. The text has been revised accordingly.

441 – earlier mention of hydrophobic and hydrophilic extractives, but only acetone used here...does that miss some compounds?

Response: acetone is a broad solvent for this purpose as it can extract most classes of extractives. It is possible that it misses a few compounds, but it is viewed as a very good compromise to leach out the majority of extractives. As mentioned in the introduction, in spruce bark most extractives have a hydrophobic character, making acetone a very good alternative. Doing sequential extractions with solvents of varying polarity is possible and has been done previously in chemical characterization of bark, but our overall goal was first of all to see *if* something happened to the extractives and if so to try to pinpoint *what*, for which acetone worked well. We have added information in the Methods to clarify this, as “The bark samples were extracted with 200 mL acetone as solvent, in order to extract the majority of extractive compounds, with 10 min per extraction for 24 cycles” on lines 474-475.

455 – how was 205 nm determined? Acid-soluble lignin from different sources determines different wavelengths selected. I would add citation here.

Response: 205 nm has been chosen from trying to avoid overlapping UV absorption from carbohydrate dehydration products formed during sulfuric acid hydrolysis. Lignin strongly absorbs UV at 205 and 280 nm, but also furfurals absorb around 270-280 nm, so at 205 nm one can assume that the contribution from other structures (mainly furfurals) is minimal. It is a very good suggestion to add a reference, which we have done in the methods (<https://doi.org/10.1515/hf-2013-0233>).

463 – hydration factors to recapitulate polymer concentrations?

Response: hydration factors could be used for this, but we think it would not add much to the story and in the worst case it could be misleading. Correctly assigning monosaccharides to polysaccharides is not highly accurate and we prefer discussing what happens to the carbohydrates overall rather than speculating on individual polymers, especially since in many cases we cannot distinguish between microbial and bark-derived carbohydrates.

465 – I see no compensation to the ‘illusion of gain’ issue when mass is removed from solid substrates. If no lignin were removed, but 100% of carbohydrates were removed, the lignin concentration would go up despite no lignin being formed. Must multiply by fraction of original.

Response: Exactly, this is why we use the ash standardization. The ash is the only material in the bark that cannot change/disappear over time, so to normalize our values compared to the ash gives a good view of how the different components change over time. If we only compare the proportions of the other components, we will not take the amounts lost as volatiles into consideration. For each sample, approximately half was frozen and sent for microbial/DNA analysis, while the rest was used in chemical analyses (after careful weighing) which made determination of total dry weight impossible.

We have clarified the experimental setup, which was also commented on by Reviewer 3, to explain why we use the ash content to be able to compare changes to the material over the course of the experiment.

Lines 118-120: “An active inoculum was prepared from unsterilized bark, and separate cultures were inoculated and collected over 24 weeks, with one fraction per sample analyzed by DNA extraction, and one fraction analyzed chemically.”

Lines 126-128: “As a portion of each sample was sent for DNA extraction and sequencing, and the ash cannot disappear from the material, the proportional ash content was used to normalize the data in the biotic samples.”

569 – DNA extraction from a lignocellulosic with this high % extractives and then using for metagenome assembly. Did quality warrant doing this? Qubit and Nanodrop, but results/QC?

Response: we were also concerned before beginning this study as to whether spruce bark would be amenable to DNA extraction and analysis, so we did some pre-tests to confirm that the protocols worked. The majority of the DNA was >5000 bp, which is recommended for amplicon sequencing, and this information has been added also to the methods. The Q-score for both Illumina and Oxford Nanopore sequencing can be found in Supplemental Table S5 and were both of good quality.

Besides concentration measurements from qubit and nanodrop, the nanodrop also provides several important QC ratios indicating presence of contamination (e.g. RNA) and we obtained good results. Furthermore, genomic DNA screentapes were used to assess size distribution of extracted DNA. Two different extractions were performed for Oxford Nanopore Technology (ONT) and Illumina, both providing similar extractions statistics. Both Illumina and ONT sequencing libraries performed well. 340 Gb of raw Illumina data with median Q-score 33.8 was produced, while 16 Gb (2.9 mio reads) long reads (median read N50 of 6500) with median Q-score 14.5 was produced with ONT. Please note that the ONT Q-score was on par with the at the time current kit chemistry and basecalling algorithms.

Results

Fig 2 – similar to fig 1, I just don’t think this is helping us much. Maybe in supplementals.

Response: this figure has been removed and a more detailed description of the experimental setup added to the end of the Introduction and beginning of the Results.

124 – Ash will be a very small percentage here, it looks from S1 like 5% (high relative to wood ash), so 53% gain means 20 mg/g or so gain. Also the S1 shows a big difference between the time zero for ‘blank’ vs. ‘sample’ that, if averaged, would make the 53% more like 21% I think.

Response: the high ash content in bark compared to wood has been clarified in the Introduction (line 53) and in the new Figure 1. Regarding the difference between the blank and the zero time point, this

comes from two separate batches of bark being used for the respective measurements, due to an unfortunate contamination of some of the original blank samples which made the replicates unreliable. We have added this information to each caption of Figures S1-3. This was a major setback with these very long incubations, but we have made sure to not discuss absolute values between blank and biotic samples but rather changes within each dataset and we would like to stress that the main point of the blank sample is to monitor whether anything happens over time from abiotic reactions, which we could not observe to any large extent. Also, to our knowledge we have not seen similar studies of sterilized bark over time. Concentrations of the measured components are within previously reported values for both bark batches.

S1 – While I am inspecting S1, I would note that a similar major disparity between time zero materials but with error bars showing significant differences is in the lignin, with non-inoculated being 30% and inoculated ‘biotic’ treatment being 40%. That, to me, would indicate problems in one of two things: 1) your acid hydrolysis technique, or 2) replication or actual, physical sample size issues leading to noise that, for some reason, is not reflected in error bars. 30% lignin vs. 40% lignin...that is very large disparity in material from same source at time zero stage.

Response: please see response above. Bark is a heterogeneous material, and we have used milled bark from industry and not from a single tree or single part of an individual tree, so there is bound to be some variation though our data therefore also represent a more general picture of spruce bark than sampling a specific tree might. Again, we would like to stress that no major changes of lignin/acid-insoluble residue could be observed within each batch over the timespan of the experiment.

137 – if you had presented the bark in its natural state, you could have simply measured mass loss and had your answer. Instead, you used ash content to normalize data to try to deal with the ‘illusion of gain’, and I’d be uncertain about conclusion here.

Response: please see the response above concerning ash content. We are not fully sure what the reviewer means when it comes to monitoring lignin content however. If we had only done chemical analyses and measured mass loss, we could have compared the lignin content to our initial values, but this quantitation is still difficult as it, still, mainly relies on gravimetric analysis after sulfuric acid hydrolysis known as Klason-lignin analysis or a variant thereof. The measurement of the insoluble residue is not necessarily easy to interpret as it could be affected by the presence of non-lignin acid-insoluble material, e.g. from microbes growing in the material.

To clarify the latter part, we have added the following text to the Discussion, on lines 362-364: “It is worth pointing out that some of the measured monosaccharides likely originate from the microorganisms, and these could not be separated from the bark itself, and possibly the acid insoluble residue could partially comprise microbial-derived molecules in addition to lignin.”

177 – I think these specifics are interesting, for sure. Looking within a fraction at loss patterns.

Response: thank you for the nice comment.

191 – Or they are inhibited in anaerobic conditions, not extractives, but as your samples dried, were able to outcompete others.

Response: please see our response above concerning aerobic/anaerobic conditions.

359 – Or your conditions were conducive to anaerobic, resin-degrading bacteria.

Response: this is a very interesting comment, as in the absence of oxygen, some resin acid biotransformation has previously been observed though no evidence exists for the degradation of the carbon skeleton (<https://doi.org/10.1007/s002030050752>). Unfortunately, there is still a great lack of

detailed information on biological resin acid degradation in literature. As mentioned previously, we do not think our conditions were anaerobic, but we have addressed this possibility in the Discussion.

Lines 391-394: “While the cultures were fully exposed to air, we cannot exclude the possibility that (micro)anaerobic pockets were also formed in the material during the experiment, though as full conversion of resin acids has not been observed anaerobically²², such micro-environments would likely be minor.”

401 – The *Pseudomonas abieticivorans* discovery is cool. I think you’ve got an interesting bacteria in hand to work with, I just find this part to be something different from the rest of the study. It struggles to link.

Response: thank you for the nice comment about our discovery. We understand that the protologue for this new type species might seem out of place, but the species description should by convention be placed in the main text. We have revised the heading to clarify that this is the protologue that stands apart from the main text, and if it is possible it would be nice to show it for instance within a box in a published version of the manuscript.

Reviewer #2 (Remarks to the Author):

Dear Authors,

thank you for opportunity to review your manuscript about degradation of resin acids. I agree that the topic of microbial bark degradation is overlooked while being relatively important for forest ecosystem and for handling of waste from wood industry. The authors performed characterization of microbial community which was found to be growing on decomposing bark over the course of 6 months, extensively described chemical changes of the substrate and isolated bacterium which is able to grow on resin acids.

Generally, I like the extensive character of the study and the level of data reporting. Manuscript contains well prepared figures, a lot of supplementary data which might be suitable for interested reader. Reproducibility seems feasible as most of the raw data are given and Zenodo archive contains full folder/file structure which was used for analysis. Long read sequencing surely improved overall quality of metagenomic and genomic datasets and made the re-analysis of data more attractive for scientific community.

I certainly understand the temptation to present all the obtained data. While this is fine in most cases, I see also part which is irrelevant in the context of the current manuscript. I feel that fungal data which are mentioned only once in discussion (diversity, 359) and did not make it into abstract might be omitted without losing any key information from the manuscript.

Despite the aim to present all the data, a few important parameters of the study are not clear. These are replication design and clustering parameters of amplicon datasets (see line by line comments). Introduction part and Methods section are well-written. Although Results and Discussion parts are

separated, several parts in Results would better fit in Discussion. Probably from this reason, the discussion is rather brief.

Overall, the presented work shows novelty in terms of researched topic while utilizing standard, widely applied techniques. The level of progress in the areas of forest ecology/carbon cycle/microbial role is moderate. Together with my opinion that the manuscript is not ready for publication in Nature Communications in its current form, I recommend its rejection.

Response: as the reviewer did not find our experimental setup very clear, we have strived to improve it in our revised version. We have addressed all the comments below, but can summarize here that the aim of our study was not to necessarily sample “natural” communities on trees in nature, but rather lay the foundation for understanding how bark can be degraded biologically by focusing on a microbial enrichment. This process is not known from previous works, and without having a solid understanding of what chemical steps constitute this degradation we could not have known what stage of degradation we were monitoring, for instance if we had done a community/metagenomics analysis and chemical characterization from a selected piece of bark. Additionally, without controlling the environment of this degradation, we could not have known whether other factors play a role in the results, and in nature these are plentiful. We believe the information presented in our manuscript is a significant step in understanding the fundamental process that spruce bark degradation represents, and will both put previous studies into context and pave the way for new studies of bark degradation in the forest environment.

Please see line by line comments and note that not all of them are negative.

59 - Bark mass estimate is worth of mentioning also here, not only in discussion.

Response: we have moved the amount figure to the introduction, as suggested, and according to a comment also from Reviewer 1. The text now reads:

Introduction, lines 58-62 “Spruce is one of the dominant tree species in northern hemisphere forests and of high economic value, and its bark comprises around 10-15% of the volume at harvest. In the Nordic countries alone, bark is an industrial side stream produced in close to 400 million m³ per year⁴², but it is largely seen as an industrial side stream today and is either left to rot in the forest or burnt for energy at the mill.”

Discussion, lines 414-418: “Of the millions of cubic meters spruce harvested annually, the bark comprises a significant fraction which today is poorly utilized; its high moisture and ash content make burning inefficient but its polymers and especially the high proportion of extractive compounds with varying properties could potentially be utilized to develop new products.”.

77, 608, 616 - There are more resin degrading genera, but *Pseudomonas* was targeted in downstream workflow by medium and primers for marker gene. Could you please elaborate on that?

Response: the reason for focusing on *Pseudomonas* was simply because of its dominance in our samples as described in the Metagenomics section, and we wanted to maximize the chances of isolating such a species and also more easily identify it.

106-112 - not fitting to Results

Response: we do not fully agree with this comment. An introductory type of text is neither inappropriate nor uncommon in the beginning of a Results section to further explain the experimental setup and so to speak set the stage for the story, but we have revised the text both in the last paragraph of the Introduction and here, to focus on the bark sourcing and sterilization which is a result in itself, and to make the experimental setup more obvious especially since we have removed the old Figure 2. We believe our long-term and dual-type analysis of both communities and chemical changes will be inspiring to the field and want to have it clearly described in the main text:

Lines 115-120: “Bark was sourced from a pulp and paper mill after having been stripped from the logs. To sterilize the bark without excessively disrupting its structure, it was gamma-irradiated instead of autoclaved, as elevated temperatures and pressure would release or modify the extractives, and doses over 20 kGy were found to be enough to stop any growth. An active inoculum was prepared from unsterilized bark, and separate cultures were inoculated and collected over 24 weeks, with one fraction per sample analyzed by DNA extraction, and one fraction analyzed chemically.”

126, Fig S2 - I do not see reason for ash standardization. At least monosaccharides are partly burnt to CO₂ which makes standardization useless. Per g of dry mass standardization is enough.

Response: there seems to be a misunderstanding of how the ash measurements were done. For ash analysis all organic matter is combusted to leave behind only minerals, that do not disappear. Therefore, we chose to use the ash content which cannot change in the material, as a proxy for degradation, and partly it will also reflect the growth of the microorganisms as bark biomass and microbial biomass cannot be separated. Please also see the responses regarding ash content to Reviewer 1.

138 - Chitinolytic bacteria feeding on fungi might also represent some "microbial degradation". This is rather side note, the sentence is fine, although a little bit bold.

Response: yes, of course microbial degradation would involve everything that they do, but we still think the sentence is not misleading and it has not been changed.

184, Table S3 - Table contains ASVs, although ASV generation is not mentioned in the Methods.

Response: thank you for pointing this out. The information on ASV generation has been added to the Methods (line 604).

185 - 194 - Are fungi indeed needed in the present manuscript?

Response: yes, our aim was to map which species can grow on the bark, where both fungi and bacteria could play important roles. That the fungi did not appear to dominate the enrichment was somewhat surprising but is important to mention this to fully describe our results. Fungi have been mentioned more in the revised manuscript:

Line 96 in Introduction and Discussion paragraph on lines 383-394.

216 - Calculation of mean and SD based on duplicates is problematic from my point of view.

Response: we have revised both main text and supplemental figures to show individual data points, and include lines to show the means as a way to illustrate trends. The captions have been updated accordingly.

224 - 229, Fig S8 - Was annotation based on metagenomic contigs or reads? Description of such annotation is missing in Methods. Annotation of eukaryotic metagenomic contigs is very problematic due to their fragmentation.

Response: the methods for the annotation has been added to the methods on lines 632-635. The annotation was based on contigs, but we have updated Figure S8 to show also the situation when basing it on reads. We agree that estimation can be difficult, but both methods verify that bacteria were dominant in the enrichment. Also in the results we have updated the text to specify that the eukaryotic content was based on contigs, on line 227.

Figure S1. Microbial taxonomic composition after two weeks of spruce bark degradation. A) Relative phyla, B) class abundances based on 16S rRNA gene target sequencing (16S) and whole metagenome (MG) reads. Comparison of fungal, bacterial and unclassified abundance in sample based on C) contigs, and D) reads.

232-235, 240-243, 257-261, 298-300, 305-309 - not fitting to Results, I think that any discussion should be avoided here since Discussion and Results are separated.

Response: we do not agree that all discussion should be removed from the Results, and rather think it is very important to have some discussion of individual results in the Results section, to improve the understanding and interpretation of the content for the reader. It is not uncommon in other publications and not a hard rule, and we do not see it as problematic to for instance focus more on the bigger picture in the Discussion while still discussing “smaller” individual results or data in the Results section. Having to revisit every result again in the Discussion in order to be able to discuss them risks making the text very staggered and difficult to follow. We do agree on some of the suggested changes.

Lines 232-235: in order to explain the Results we need to provide some background information here, and it is not fitting to the Introduction before we even come to these individual results. It would not make sense to introduce details on resin acid degradation before these results actually are presented. It is not inappropriate to bring up other studies to explain the actual research conducted and simply starting to talk about a dit cluster analysis directly would not make sense.

240-243: the second part of the sentence has been moved to the Discussion, as this fits with a bigger scope that we are aiming for. Lines 404-407: “The observation that other MAGs did not encode full dit clusters might indicate that these organisms have incomplete resin acid degradation pathways, unknown pathways, or are in fact unable to fully metabolize resin acids.”

257-261, and 298-300: we have not changed the text, with a similar reasoning as above. These results if reported only as a result without any interpretation would seem out of place in our eyes.

305-309: we have not changed the text, but added to the Discussion to recapitulate this finding, lines 402-404: “Our metagenomic analysis also facilitated the discovery and subsequent characterization of *P. abieticivorans*, which dominated our enrichment cultures and encodes a complete *dit* cluster while appearing to have a limited capacity for complex carbohydrate metabolism.”

237 - this is nice finding!

Response: thank you.

285 - MAG15 with 69 contigs are used as proxy for abundance of identical isolate PIA16. It would be interesting to assess the abundance of PIA16 itself by mapping metagenomic reads to genome. It is complete bacterial genome, one contig, the best representation of the target organism.

Response: unless we are misunderstanding the comment, this is exactly what we did, and found PIA16 to have an abundance of 19.2% (line 239-240) in the metagenome (=MAG15), which has been clarified by adding contigs. If referring to mapping reads instead of contigs, the abundance of PIA16 in the metagenome was estimated to 6.88%.

296, 665 and Table S11 - Are there reference papers for other five resin acid degrading bacteria? I recommend to include them.

Response: these have been included.

377 - Inverse relationship can be tested, right? I suggest to perform such test.

Response: this was not meant as a firm statement as it is part of the Discussion, but rather raising the point that the most abundant MAGs appear to encode fewer degradative CAZymes than some of the MAGs of lower abundance, and this could relate to the preferred metabolism of extractives rather than carbohydrates. The text has been modified to lessen this statement and now reads:

Lines 398-400: “Interestingly, the number of encoded degradative CAZymes in our reconstructed MAGs was relatively low in the MAGs of highest abundance (Fig. 4 & Fig. S11), which might explain the observed low carbohydrate turnover.”

400 - After successful publication, please consider to send description of type strain to the staff of IJSEM in order to publish the new name also effectively.

Response: Yes, this will definitely be one of our top priorities.

417 - Please consider to use superscript "T" for type strain throughout the manuscript.

Response: good suggestion. We have added the superscript T to strain names when mentioned and if they are type strains.

423 - Is it possible to note the age of source trees for bark? Is it important at all?

Response: it is not possible to know the exact age of the trees, but the average age of harvest is 70 years in Sweden, and as we used bark from an industrial pile our assumption is that the average age of also our bark samples lies around this number. Please see the answer below also for line 435.

429 - Just to clarify - unsterilized AND milled bark was used, right? What were the temperature and light conditions during inoculum growth?

Response: yes, the bark was milled but not sterilized. The temperature was ~21 °C and the light conditions 16 h light and 8 h dark per day cycle. The information has been added.

434 - What were the light conditions?

Response: 16 h light and 8 h dark per day cycle. The information has been added.

435 - This is one of my main concerns. Please clarify "biological triplicate experiments". I imagine that biological replicates represent bark from more than one experimental site or bark of differing age or differing culturing conditions or another grouping. If the bark came from one batch and cultivations were identical then all cultivations are technical replicates. This is relevant also for line 512 - DNA extractions where duplicates are not mentioned. SRA records show duplicated datasets for each sample and for each marker gene. Fig 3 and Fig 5 captions mention biological duplicates. Fig S1 and Fig S2 mention biological duplicates and two technical replicates. I found the replication description confusing, could you please clarify it?

Response: the bark came from a pulp and paper mill, where the procedure is to strip the spruce logs at the mill after harvest and gathering the bark in large piles. We used a large amount of bark from such a pile, after cleaning off all visible wood fragments, before milling it. Our bark samples thus represent an average bark composition from many trees aged around 70 years (standard harvesting age). The biological triplicate experiments refer to each sample of bark (10 g) being inoculated with the same inoculum and three samples started per timepoint, though duplicate analyses were performed for each after collection, essentially due to the excessive resources needed to extract and in detail analyze each and every sample. The small variations we show in our results indicate that this was a successful experimental setup to follow the progress of both the enrichment cultures and the chemical degradation. We would however not call these cultures technical replicates, analogous to how separate cultures of e.g. *S. cerevisiae* are regarded as biological replicates while subsequent analyses can be done in technical replicates.

The bark collection has been updated to “Spruce bark was obtained from the Iggesund pulp and paper mill (Iggesund, Holmen AB, Sweden), from a bark pile resulting from stripping of spruce logs at the mill after harvest, with the average age of trees at harvest being ~70 years” on lines 448-450.

The DNA section has been updated to “For each timepoint, duplicate spruce bark samples were subjected to DNA extraction and community analysis performed by DNAsense ApS (Denmark), ...” on lines 546-550.

554 - Could you please specify if any ITS extraction method was used?

Response: thank you for spotting this. We have added a new subheading to have one called “Bioinformatic processing for bacterial gene regions” and one “Bioinformatic processing for fungal gene regions”.

566 - Could you please specify metagenome sampling timepoint here? It appears in Fig 2 caption and on line 208.

Response: this has been added.

578 - Probably, this is not the case but based on current sentence it seems that fragmented DNA was used also for long read sequencing. Please clarify.

Response: thank you for noticing this. We have clarified which methods were used for Oxford Nanopore sequencing and which for Illumina sequencing.

582 - Out of curiosity, could you please explain reason for subsampling raw metagenomic data?

Response: The reason for subsampling data above 100 Gb is due to computational feasibility, and the subsampling has minimal impact on the sensitivity of the analysis.

604 - Which version of dbCAN was used? I suggest to report it.

Response: we used dbCAN2, and have updated the manuscript accordingly.

738 - minor issues with data deposition, deposition itself is good - in order to be consistent with NCBI accessions, I recommend to use "Run" accession for SRX18691017 and SRX18691016 (SRR22729505 and SRR22729506, respectively). SRR22745721-SRR22745728 are not described in this paragraph although they are present in BioProject, these are probably ITS data. SRR23100117 is metagenomic dataset without further description, could you please add it? Could you please deposit also raw genomic data of *P. abieticivorans* PIA16?

Response: thank you for pointing this out. The text has been updated in accordance to this on lines 782-788, as seen below. Raw genomic data for *P. abieticivorans* PIA16 are now deposited at NCBI with BioProject ID PRJNA803013.

“All sequencing reads have been deposited at the National Center for Biotechnology Information (NCBI) under BioProject ID PRJNA912085 and PRJNA803013. The 16S, ITS, PIA16 and metagenomic reads have been deposited in the Sequence Read Archive (SRA) under the accession numbers SRR22745541-SRR22745556 (biosample accession SAMN32240041-SAMN32240056), SRR22745721-SRR22745736 (biosample accession SAMN32240041-SAMN32240056), SRR24684300, SRR24684301 (BioProject accession PRJNA803013, biosample accession SAMN25609858) and SRR22729505, SRR22729506 (biosample accession SAMN32218794) respectively (Table S13).”

We have also added a description to SRR23100117, as the “Metagenomic sequencing of a bark degradation timeseries experiment”

Fig 5 - shows detailed analysis of gene content, I like this representation of data, it shows the depth to which authors analyzed data.

Response: thank you very much for the nice comment.

Reviewer #3 (Remarks to the Author):

In manuscript, entitled “Resin acids play key roles in shaping microbial communities during degradation of spruce bark”, the authors claim to have analysed key stages in degradation of bark, an abundant renewable resource, and how bark defensive extractive compounds have major roles in shaping microbiomes.

While contributing with important information about the evolution of certain compounds of the bark along the development of certain inoculated microbial strains, which were previously artificially cultivated, the work misses the opportunity to analyze how bark is truly degraded in nature by natural microbial communities and how this natural communities truly evolve.

I have two major concerns:

1. To set up the experiment, the authors produce a culture of microbes which proceed from bark cultivated in M9 medium for 6 months. I wonder why they do this. When bark is moved from its original environment and it is cultivated under laboratory conditions, at certain specific temperature (not fluctuating), with no day/night variations, the microbial communities are going to completely change compared to the natural ones. Moreover, it seems by the figure 2 (even if not included in the methods) that the authors use agar for this cultivation, which can be used as solely nutrient provision for many microbes, so it even modifies more the microbial communities originally present in the bark. During the culturing period, and at those different conditions, some of the taxa are going to overgrow, and some of them can produce and excrete different molecules that can inhibit other microorganisms or, by the contrary, to promote them. Also, some serendipitous microbes, which could even be as spores over the bark and maybe not developing in natural conditions, can grow in those artificial ones.

Therefore, the inoculated microbial community is going to be completely different to the originally present in nature, and that will impact not only the analysis of microbial communities, but also the degradation of bark compounds, compared to what it happens in nature. In other words, the way that the experiment was set implies that no matter if the original source to obtain the inoculum is bark or any other source, because the inoculated community is going to be completely different to the natural one.

For these reasons, I consider that the authors did not choose the best approach for their analyses. Why didn't the authors just evaluate the naturally present communities and analyze them and the extractives over the time?

Response: we have tried to clarify in the revised manuscript that our intention with the study was to follow how bark can be degraded by microbial consortia, and importantly over time. As we state in the Introduction, how this process looks/proceeds is not known today. If we had analyzed different bark samples in nature we would not have gotten the answers as in this controlled experimental setup, where key questions were: how is the bark degraded chemically, and which types of microorganisms may be involved? To "just evaluate naturally present communities" is of course interesting, but without having the basal understanding of how the bark is degraded over time chemically, we cannot say what they are doing and we need to *both* have a controlled bark sample *and* be able to monitor this over time, both chemically and community-wise.

Of course our initial setup of a consortium is artificial in a way, even though it comes from recently harvested trees, but as we also mention in the Introduction, former studies of bark degradation indicate this being a very slow process where even after waiting years to sample it seems to still be ongoing (with the caveat that as we say, the knowledge of what would be a chemical marker for what is the degradation stage is not known) and the risk of losing key species we see as low.

We thus wanted to make sure we would actually observe growth on the bark during a reasonable time frame, and secondly to be able to repeatedly measure what happens to it during said degradation. Had we taken an initial inoculum directly from a tree we might not have observed growth during the 6-month time frame and abandoned the experiments a few months in. Had we taken it from an old bark pile, or soil, the study would still have been biased in one way or another as choice of inoculum will always be contextual, and one can always argue that an inoculum is artificial – from which tree/forest/soil/year – all may have an influence on the outcome. We do believe that our study will provide a key starting point for other comparative studies with different inocula to see if the degradation always follows a similar pattern with (especially initial) domination by Pseudomonadota (Proteobacteria) or if it is highly variable.

Sampling in nature and doing chemical analyses coupled to community analyses is highly interesting, but monitoring the degradation stage is not possible without a clear beginning and end as well as the stages in between, which before our study were unknown apart from the beginning step based on chemical analyses of fresh bark. As commented on below, regarding missing literature, we have found no similar studies that would have guided us to know exactly which samples to further study by DNA analyses, and sampling various trees, fallen logs, or bark samples would not enable us to piece together the degradation process in an easy-to-follow manner.

Regarding figure 2, agar was not used. The figure has been removed, but we have supplied a picture of the bark to Reviewer 1, above, to show what the actual bark samples looked like before inoculation.

Clarification of our experimental setup has been added as follows:

Lines 88-92: “In order to understand how bark can be degraded, the process needs to be monitored over time and both microbial and chemical analyses need to be combined, to answer the questions of what type of species are involved or dominate and what bark compounds are mainly affected. Here, we have investigated spruce bark degradation over six months using an enriched microbial community sourced from an industrial bark pile, and combined several methods to obtain a broad picture of the entire process.”

Lines 115-120: “Bark was sourced from a pulp and paper mill after having been stripped from the logs. To sterilize the bark without excessively disrupting its structure, it was gamma-irradiated instead of autoclaved, as elevated temperatures and pressure would release or modify the extractives, and doses over 20 kGy were found to be enough to stop any growth. An active inoculum was prepared from unsterilized bark, and separate cultures were inoculated and collected over 24 weeks, with one fraction per sample analyzed by DNA extraction, and one fraction analyzed chemically.”

2. Regarding the description of a new species, the closest related type strain to PIA16 according the sequence of the 16S rRNA gene is *Pseudomonas laurylsulfatiphila* DSM 105097 (16S rRNA gene: NIRS01000004). Both strains share 99.18% similarity in this gene. As far as I can see (the phylogenetic gene is huge and has a very small letter size), this species has not been considered for the phylogeny -and it makes not sense to include such a big number of unrelated species in the tree-. Anyways, it has not been estimated the ANI value between PIA16 and *Pseudomonas laurylsulfatiphila* DSM 105097. The calculation of the ANI value with this strain is required in order to decide if PIA16 is a new species, since the similarity in the 16S with this strain is much higher than that with the type strains considered in the MS, and also it should be included in the phylogenetic analysis.

Response: the phylogenetic tree showed the results of the analysis performed comparing to the rRNA/ITS database which NCBI describes as "This set is critical for correctly identifying and classifying prokaryotic (bacteria and archaea) and fungal samples" (<https://ncbiinsights.ncbi.nlm.nih.gov/2020/02/21/rrna-databases/>). As result, it did not show any result for *Pseudomonas laurylsulfatiphila* DSM 105097.

However, we have remade the phylogenetic tree to make it more focused on the most closely related species based on 16S sequence similarities, but also including the known resin acid-degrading *Pseudomonas* species (Figure S20, see below). The phenotypic characteristics of *Pseudomonas laurylsulfatiphila* DSM 105097 have been added to Table S10, and it includes the ANI value compared to *P. abieticivorans* sp. nov, and at 81.9% it is also well below the species cutoff at 95%. Table S10 has been updated to include the most closely related species based on 16S similarity.

Other comments:

Line 58: “Spruce is one of the dominant species....in the northern hemisphere” TREE species? northern hemisphere FORESTS?

Response: the text has been revised also from a comment by reviewer 1, lines 58-59.

Line 63: Reference 6, Peng & Roberts test toxicity of bark extractives in Metazoa, I believe that it is not a good reference for microbial inhibition. Instead, there are other good articles referring antimicrobial potential of bark extractives.

Response: we agree that the reference is not optimal for antimicrobial activity but that the general toxicity of these acids is important information. We have updated the sentence to also include a new reference (Savluchinske-Feio et al. 2006) to read:

Lines 66-67-: “Resin acids in particular are known to be toxic to microorganisms⁷ as well as water-living animals with LC₅₀ values (concentration lethal to half of a population) of sub-mg/L⁸.”

Line 67-68: “Moreover, removal of extractive compounds from softwood bark has been shown to enhance its saccharification”. I do not see the point of this sentence/information in here.

Response: this sentence has been removed (see also comment by Reviewer 1).

Lines 68-70: Some important works related to bark degradation by microbes (not only fungi, but also bacteria) have been omitted here. The consideration of these studies is important in order to really see to which extend your work add knowledge.

I can refer a few of them, but there are also several others:

-Malik, R.J., Trexler, R.V., Eissenstat, D.M. et al. Bark decomposition in white oak soil outperforms eastern hemlock soil, while bark type leads to consistent changes in soil microbial composition. *Biogeochemistry* 150, 329–343 (2020). <https://doi.org/10.1007/s10533-020-00701-7>

-Hagge Jonas, Bässler Claus, Gruppe Axel, Hoppe Björn, Kellner Harald, Krah Franz-Sebastian, Müller Jörg, Seibold Sebastian, Stengel Elisa and Thorn Simon 2019Bark coverage shifts assembly processes of microbial decomposer communities in dead wood *Proc. R. Soc.* B.2862019174420191744 <http://doi.org/10.1098/rspb.2019.1744>

-Dong Wang, Olatunji Olusanya Abiodun, Jinlan Xiao, Wenqiang Zhao, Contrasting responses of microbial diversity and community structure in decaying root bark and xylem to N addition in an alpine shrubland, *Soil Biology and Biochemistry*, Volume 178, 2023, 108937, ISSN 0038-0717, <https://doi.org/10.1016/j.soilbio.2022.108937>.

-Weedon, J. T., Cornwell, W. K., Cornelissen, J. H. C., Zanne, A. E., Wirth, C., & Coomes, D. A. (2009). Global meta-analysis of wood decomposition rates: A role for trait variation among tree species? *Ecology Letters*, 12, 45–56. <https://doi.org/10.1111/j.1461-0248.2008.01259.x>

Response: thank you for this comment, and we agree we could put our experimental setup and results into context in a better way. We have included some of these references and others in the revised manuscript to highlight both the breadth of our study and how it relates to other work. As a brief response, in none of the articles we have found there is a similar holistic study of bark degradation and the temporal aspects of this process is unclear, whether on chemical or species level. For instance, the references mentioned here by the reviewer either concern bark or bark+wood (root) samples that were buried in soil (Malik et al. 2020 and Wang et al. 2023) which is not comparable to our study. In none of the listed references did the authors report detailed chemical analysis of the bark or following the degradation over time but rather single time points, and we have not found such studies in literature. The study by Hagge et al. 2019 is of course quite relevant as it concerns spruce, but they sampled much more wood than bark (full end-to-end cores drilled out), their sampling was done on logs after 1.5 years of felling a tree and without chemical analysis, which again is not directly comparable.

The manuscript text has been revised as follows:

Lines 70-74: “Snapshot studies of which microbial communities can colonize bark of different trees have been conducted, for instance analyzing bark remaining on logs or samples buried in soil after years of incubation¹⁰⁻¹². However, how these communities correlate to degradation stage is currently unclear, as simultaneous detailed chemical analyses of the bark have not been conducted over time.”

Lines 388-391: “However, in some studies we see similar profiles of microbial abundance as reported here, with a dominance of Pseudomonadota for bacteria¹¹ and mainly basidiomycetes and ascomycetes for fungi^{11,12}, though without chemical analyses of the bark itself knowing what stage of degradation these previous studies correspond to is not possible.”

Line 131: “However, a larger effect could be seen for glucose (Fig. S1 & Fig. S2)” Please, slightly comment this effect (do not only refer to the figures). Also related to this: How do you explain the

increase in glucose content within the uninoculated control over the time? (Figures S1 and S2). Once you make your study considering an initially sterilized bark, have you tested if the bark was really sterile? Is the gamma radiation killing microbes inside the bark? This should be tested. A possible presence of microbes in the negative controls could be the reason of the glucose increase over the time.

Response: regarding glucose, the text has been revised to say “However, a larger effect could be seen for glucose (Fig. S1 & Fig. S2), where a slow but continuous decrease was observed which could be attributed to cellulose- or perhaps more likely starch degradation, ...” (lines 133-135). We cannot fully explain why the glucose appears to increase in the final blank sample, as glycosidic bonds at least within poly- or oligosaccharides are not prone to spontaneous hydrolysis. We do study the process over 6 months, so it is possible that the apparent released glucose could come from more labile glucose-containing molecules such as stilbene glucosides. To discuss this in detail we however think would be too speculative.

We have added on lines 136-138 the following sentence: “In the blank sample, an apparent increase in glucose was observed in the 24-week sample, possibly stemming from less stable glycoconjugates than polysaccharides.”

Regarding the gamma irradiation, yes, since the rays fully penetrate the material they also reach microbes within the bark. We did perform the necessary tests to verify that the bark had been properly sterilized and compared 0, 10, 20, and 30 kGy doses of the bark before adding to rich media (LB and PDA). No growth could be seen after the bark had been exposed to 20 kGy, after careful observation for more than a week’s incubation (see below). We then chose 25 kGy to be fully sure to avoid contamination. We have included this information on lines 115-118 and 452-455.

		kGy			
		0	10	20	30
positive control	+				LB					Growth		+	+	+	-
PDA					Growth		+	+	-	-

Lines 133-134 and lines 144-145 are discussion, not results.

Response: the sentences have been moved to the Discussion.

Line 235: The presence of some *dit* genes within distant MAGs could indicate some event of horizontal transfer to fulfil the adaptation to the bark environment. However, the presence of incomplete (no-functional) *dit* clusters would not be so advantageous and could be just a transient gene gain. Have you compared the sequence of the *dit* genes among MAGs to see whether they are divergent or not? Have you searched for transposable elements within these contigs? Is the relative abundance of the *dit* genes concordant with the relative abundance of each MAG? These issues could help to gain a better perspective of the dynamics of these genes within the microbial community. This paper exemplify a similar issue based on another gene cluster:

<https://www.nature.com/articles/s41467-018-04955-6>

Response: This is a very interesting question, and we agree it is a good idea to go deeper into detail and explain the relationships between the clusters a bit more. We need to first stress that resin acid degradation is poorly understood today. From what is known, not all genes within the *dit* cluster are essential for resin acid degradation, meaning that we cannot not know if what we see as incomplete *dit* clusters are non-functional or simply encode the bare minimum of genes needed for resin acid degradation and/or degradation of other aromatic compounds. However, the *ditA1* gene/DitA1 protein, has been found to be essential for degradation of resin acids in both *Pseudomonas* and *Paraburkholderia* species and the presence of homologues appears to be prevalent among organisms that can degrade aromatic hydrocarbons. Moreover, it has been reported (Witzig et al. 2007, <https://doi.org/10.1111/j.1462-2920.2007.01242.x>) that the DitA1 gene was not transferred through horizontal gene transfer, but was present in the most recent common ancestor of *Pseudomonas*, *Cupriavidus* and *Burkholderia* before speciation occurred and evolved following this event.

We have not compared all *dit* genes among the MAGs, but we do think that this is an important point and have compared all *ditA1* genes against each other. The results from this can be seen in the table below the graphs, and in the vast majority of cases the sequence identities were around 30% which does not suggest a recent horizontal transfer events. The few instances of higher identity are apparent recent duplications within the same species and one case between MAG14 and MAG15. We used a cutoff E-value of 1e-10, and no cutoff for sequence identity.

We have added the following text to the Results, including the Witzig et al. 2007 reference:

Lines 243-246: “Comparing all *ditA1* homologs among the species showed sequence identities around 30% for the vast majority, which corroborates previous studies suggesting that *ditA1* is not predominantly shared via horizontal gene transfer²⁶.”

Regarding the relative abundance of *dit* genes, we cannot say for certain that there is a correlation (see below). Given the paucity of knowledge regarding the function of all *dit* gene products, we would like to refrain from speculating too much about such correlations and what they could mean.

Table of sequence identity between each *ditA1* gene found in the MAGs.

Query	Hit	Seq. Id (%)	E-value
MDE1173575.1 MAG4	MDE1173575.1 MAG4	100.000	0.0
MDE1173575.1 MAG4	MDE1197284.1 MAG14	48.900	4.90e-147
MDE1173575.1 MAG4	MDE1159551.1 MAG8	45.564	5.17e-125
MDE1173575.1 MAG4	MDE1167041.1 MAG15	43.276	1.37e-119
MDE1173575.1 MAG4	MDE1197273.1 MAG14	41.769	6.44e-113
MDE1173575.1 MAG4	MDE1174304.1 MAG4	28.960	4.74e-40
MDE1173575.1 MAG4	MDE1165299.1 MAG15	31.658	3.87e-27
MDE1173575.1 MAG4	MDE1169134.1 MAG15	29.947	3.19e-25
MDE1173575.1 MAG4	MDE1181026.1 MAG7	27.556	1.27e-23
MDE1173575.1 MAG4	MDE1180490.1 MAG7	27.556	6.23e-23
MDE1173575.1 MAG4	MDE1168779.1 MAG15	27.189	5.23e-22
MDE1173575.1 MAG4	MDE1186864.1 MAG9	27.723	2.39e-20
MDE1173575.1 MAG4	MDE1172124.1 MAG4	24.384	5.76e-20
MDE1173575.1 MAG4	MDE1141657.1 MAG6	32.857	9.31e-19
MDE1173575.1 MAG4	MDE1164404.1 MAG15	33.766	4.61e-18
MDE1174304.1 MAG4	MDE1174304.1 MAG4	100.000	0.0
MDE1174304.1 MAG4	MDE1172124.1 MAG4	29.974	7.05e-44
MDE1174304.1 MAG4	MDE1167041.1 MAG15	35.407	6.01e-43
MDE1174304.1 MAG4	MDE1197273.1 MAG14	35.238	8.52e-41
MDE1174304.1 MAG4	MDE1173575.1 MAG4	28.960	4.37e-40
MDE1174304.1 MAG4	MDE1159551.1 MAG8	34.804	3.16e-36
MDE1174304.1 MAG4	MDE1169134.1 MAG15	33.171	4.40e-35
MDE1174304.1 MAG4	MDE1165299.1 MAG15	30.091	3.64e-33
MDE1174304.1 MAG4	MDE1186864.1 MAG9	34.483	1.04e-30
MDE1174304.1 MAG4	MDE1197284.1 MAG14	33.816	1.97e-30
MDE1174304.1 MAG4	MDE1164404.1 MAG15	31.604	1.20e-28
MDE1174304.1 MAG4	MDE1180490.1 MAG7	30.374	1.71e-27
MDE1174304.1 MAG4	MDE1181026.1 MAG7	29.439	4.47e-26
MDE1174304.1 MAG4	MDE1141657.1 MAG6	29.534	2.26e-23
MDE1172124.1 MAG4	MDE1172124.1 MAG4	100.000	0.0
MDE1172124.1 MAG4	MDE1174304.1 MAG4	29.974	1.15e-43
MDE1172124.1 MAG4	MDE1186864.1 MAG9	37.019	7.29e-38
MDE1172124.1 MAG4	MDE1197273.1 MAG14	35.000	1.35e-35
MDE1172124.1 MAG4	MDE1167041.1 MAG15	34.171	2.51e-34
MDE1172124.1 MAG4	MDE1159551.1 MAG8	36.548	4.03e-31
MDE1172124.1 MAG4	MDE1141657.1 MAG6	27.397	1.20e-25
MDE1172124.1 MAG4	MDE1197284.1 MAG14	30.952	1.59e-23
MDE1172124.1 MAG4	MDE1168779.1 MAG15	28.571	6.26e-23
MDE1172124.1 MAG4	MDE1181026.1 MAG7	31.905	7.29e-23
MDE1172124.1 MAG4	MDE1165299.1 MAG15	28.837	1.06e-22
MDE1172124.1 MAG4	MDE1180490.1 MAG7	30.000	2.32e-20

MDE1172124.1 MAG4	MDE1173575.1 MAG4	24.384	8.67e-20
MDE1172124.1 MAG4	MDE1164404.1 MAG15	26.778	1.85e-15
MDE1159551.1 MAG8	MDE1159551.1 MAG8	100.000	0.0
MDE1159551.1 MAG8	MDE1197273.1 MAG14	46.985	5.83e-131
MDE1159551.1 MAG8	MDE1167041.1 MAG15	46.173	1.22e-130
MDE1159551.1 MAG8	MDE1173575.1 MAG4	45.564	5.12e-125
MDE1159551.1 MAG8	MDE1197284.1 MAG14	36.250	8.98e-92
MDE1159551.1 MAG8	MDE1174304.1 MAG4	34.804	3.40e-36
MDE1159551.1 MAG8	MDE1172124.1 MAG4	36.548	2.65e-31
MDE1159551.1 MAG8	MDE1169134.1 MAG15	34.225	8.41e-31
MDE1159551.1 MAG8	MDE1165299.1 MAG15	28.155	1.29e-29
MDE1159551.1 MAG8	MDE1181026.1 MAG7	30.317	4.00e-27
MDE1159551.1 MAG8	MDE1180490.1 MAG7	29.412	3.05e-26
MDE1159551.1 MAG8	MDE1168779.1 MAG15	26.887	1.19e-25
MDE1159551.1 MAG8	MDE1186864.1 MAG9	29.545	1.09e-24
MDE1159551.1 MAG8	MDE1141657.1 MAG6	32.867	8.59e-23
MDE1159551.1 MAG8	MDE1164404.1 MAG15	30.657	9.50e-20
MDE1141657.1 MAG6	MDE1141657.1 MAG6	100.000	0.0
MDE1141657.1 MAG6	MDE1180490.1 MAG7	44.037	3.59e-130
MDE1141657.1 MAG6	MDE1181026.1 MAG7	43.981	8.65e-130
MDE1141657.1 MAG6	MDE1168779.1 MAG15	30.597	2.10e-57
MDE1141657.1 MAG6	MDE1169134.1 MAG15	26.437	1.27e-52
MDE1141657.1 MAG6	MDE1165299.1 MAG15	30.127	3.73e-50
MDE1141657.1 MAG6	MDE1197273.1 MAG14	26.147	3.48e-27
MDE1141657.1 MAG6	MDE1167041.1 MAG15	25.571	8.83e-26
MDE1141657.1 MAG6	MDE1172124.1 MAG4	27.397	9.86e-26
MDE1141657.1 MAG6	MDE1174304.1 MAG4	29.534	3.01e-23
MDE1141657.1 MAG6	MDE1159551.1 MAG8	32.867	8.90e-23
MDE1141657.1 MAG6	MDE1197284.1 MAG14	26.263	1.90e-21
MDE1141657.1 MAG6	MDE1186864.1 MAG9	25.366	1.16e-20
MDE1141657.1 MAG6	MDE1173575.1 MAG4	32.857	9.56e-19
MDE1181026.1 MAG7	MDE1181026.1 MAG7	100.000	0.0
MDE1181026.1 MAG7	MDE1180490.1 MAG7	93.636	0.0
MDE1181026.1 MAG7	MDE1141657.1 MAG6	43.981	1.19e-129
MDE1181026.1 MAG7	MDE1169134.1 MAG15	29.561	4.59e-53
MDE1181026.1 MAG7	MDE1168779.1 MAG15	30.602	3.43e-47
MDE1181026.1 MAG7	MDE1165299.1 MAG15	27.069	1.47e-41
MDE1181026.1 MAG7	MDE1164404.1 MAG15	35.047	1.35e-37
MDE1181026.1 MAG7	MDE1197273.1 MAG14	30.233	1.45e-29
MDE1181026.1 MAG7	MDE1159551.1 MAG8	30.317	4.26e-27
MDE1181026.1 MAG7	MDE1174304.1 MAG4	29.439	5.13e-26
MDE1181026.1 MAG7	MDE1167041.1 MAG15	27.442	2.39e-25
MDE1181026.1 MAG7	MDE1173575.1 MAG4	27.556	1.34e-23
MDE1181026.1 MAG7	MDE1172124.1 MAG4	31.905	5.11e-23
MDE1181026.1 MAG7	MDE1186864.1 MAG9	27.805	8.25e-23
MDE1181026.1 MAG7	MDE1197284.1 MAG14	28.283	7.96e-19
MDE1180490.1 MAG7	MDE1180490.1 MAG7	100.000	0.0
MDE1180490.1 MAG7	MDE1181026.1 MAG7	93.636	0.0
MDE1180490.1 MAG7	MDE1141657.1 MAG6	44.213	1.27e-129
MDE1180490.1 MAG7	MDE1169134.1 MAG15	30.485	1.04e-53
MDE1180490.1 MAG7	MDE1168779.1 MAG15	29.787	1.37e-45
MDE1180490.1 MAG7	MDE1165299.1 MAG15	26.622	1.51e-41
MDE1180490.1 MAG7	MDE1164404.1 MAG15	33.178	4.49e-35
MDE1180490.1 MAG7	MDE1197273.1 MAG14	28.372	1.26e-27
MDE1180490.1 MAG7	MDE1174304.1 MAG4	30.374	1.96e-27
MDE1180490.1 MAG7	MDE1159551.1 MAG8	29.412	3.25e-26
MDE1180490.1 MAG7	MDE1167041.1 MAG15	26.047	2.59e-24
MDE1180490.1 MAG7	MDE1173575.1 MAG4	27.556	6.58e-23

MDE1180490.1 MAG7	MDE1186864.1 MAG9	27.184	4.95e-22
MDE1180490.1 MAG7	MDE1172124.1 MAG4	30.000	1.63e-20
MDE1180490.1 MAG7	MDE1197284.1 MAG14	29.592	5.25e-20
MDE1186864.1 MAG9	MDE1186864.1 MAG9	100.000	0.0
MDE1186864.1 MAG9	MDE1172124.1 MAG4	37.019	4.87e-38
MDE1186864.1 MAG9	MDE1174304.1 MAG4	34.483	1.14e-30
MDE1186864.1 MAG9	MDE1169134.1 MAG15	32.212	1.88e-30
MDE1186864.1 MAG9	MDE1165299.1 MAG15	31.963	3.28e-28
MDE1186864.1 MAG9	MDE1197273.1 MAG14	30.508	3.48e-27
MDE1186864.1 MAG9	MDE1168779.1 MAG15	31.429	4.13e-27
MDE1186864.1 MAG9	MDE1167041.1 MAG15	30.638	2.53e-25
MDE1186864.1 MAG9	MDE1159551.1 MAG8	29.545	1.11e-24
MDE1186864.1 MAG9	MDE1181026.1 MAG7	27.805	7.86e-23
MDE1186864.1 MAG9	MDE1180490.1 MAG7	27.184	4.72e-22
MDE1186864.1 MAG9	MDE1164404.1 MAG15	30.151	1.42e-21
MDE1186864.1 MAG9	MDE1141657.1 MAG6	25.366	8.67e-21
MDE1186864.1 MAG9	MDE1173575.1 MAG4	27.723	2.41e-20
MDE1186864.1 MAG9	MDE1197284.1 MAG14	27.451	5.68e-19
MDE1197284.1 MAG14	MDE1197284.1 MAG14	100.000	0.0
MDE1197284.1 MAG14	MDE1173575.1 MAG4	48.900	4.90e-147
MDE1197284.1 MAG14	MDE1167041.1 MAG15	39.312	3.12e-98
MDE1197284.1 MAG14	MDE1197273.1 MAG14	38.765	7.87e-98
MDE1197284.1 MAG14	MDE1159551.1 MAG8	36.250	9.06e-92
MDE1197284.1 MAG14	MDE1174304.1 MAG4	33.816	2.14e-30
MDE1197284.1 MAG14	MDE1169134.1 MAG15	31.034	3.83e-27
MDE1197284.1 MAG14	MDE1172124.1 MAG4	30.952	1.05e-23
MDE1197284.1 MAG14	MDE1165299.1 MAG15	27.586	4.71e-23
MDE1197284.1 MAG14	MDE1141657.1 MAG6	26.263	1.23e-21
MDE1197284.1 MAG14	MDE1180490.1 MAG7	29.592	4.98e-20
MDE1197284.1 MAG14	MDE1168779.1 MAG15	27.273	8.05e-20
MDE1197284.1 MAG14	MDE1186864.1 MAG9	27.451	5.65e-19
MDE1197284.1 MAG14	MDE1181026.1 MAG7	28.283	7.54e-19
MDE1197284.1 MAG14	MDE1164404.1 MAG15	24.737	2.66e-15
MDE1197273.1 MAG14	MDE1197273.1 MAG14	100.000	0.0
MDE1197273.1 MAG14	MDE1167041.1 MAG15	89.767	0.0
MDE1197273.1 MAG14	MDE1159551.1 MAG8	47.487	8.14e-132
MDE1197273.1 MAG14	MDE1173575.1 MAG4	41.769	2.56e-117
MDE1197273.1 MAG14	MDE1197284.1 MAG14	39.109	1.52e-101
MDE1197273.1 MAG14	MDE1174304.1 MAG4	35.238	9.18e-41
MDE1197273.1 MAG14	MDE1172124.1 MAG4	35.000	7.06e-36
MDE1197273.1 MAG14	MDE1169134.1 MAG15	31.156	1.36e-31
MDE1197273.1 MAG14	MDE1168779.1 MAG15	29.167	7.99e-30
MDE1197273.1 MAG14	MDE1181026.1 MAG7	30.233	1.20e-29
MDE1197273.1 MAG14	MDE1165299.1 MAG15	29.717	2.32e-29
MDE1197273.1 MAG14	MDE1180490.1 MAG7	28.372	9.11e-28
MDE1197273.1 MAG14	MDE1141657.1 MAG6	26.471	2.54e-27
MDE1197273.1 MAG14	MDE1186864.1 MAG9	30.508	3.69e-27
MDE1197273.1 MAG14	MDE1164404.1 MAG15	31.818	1.50e-24
MDE1164404.1 MAG15	MDE1164404.1 MAG15	100.000	0.0
MDE1164404.1 MAG15	MDE1181026.1 MAG7	35.047	1.42e-37
MDE1164404.1 MAG15	MDE1141657.1 MAG6	26.619	1.60e-37
MDE1164404.1 MAG15	MDE1168779.1 MAG15	25.604	2.36e-36
MDE1164404.1 MAG15	MDE1180490.1 MAG7	33.178	4.74e-35
MDE1164404.1 MAG15	MDE1165299.1 MAG15	34.010	5.39e-35
MDE1164404.1 MAG15	MDE1169134.1 MAG15	32.227	2.53e-31
MDE1164404.1 MAG15	MDE1174304.1 MAG4	31.604	1.46e-28
MDE1164404.1 MAG15	MDE1167041.1 MAG15	30.811	8.13e-26
MDE1164404.1 MAG15	MDE1197273.1 MAG14	31.818	1.62e-24

MDE1164404.1 MAG15	MDE1186864.1 MAG9	30.151	1.57e-21
MDE1164404.1 MAG15	MDE1159551.1 MAG8	30.657	1.07e-19
MDE1164404.1 MAG15	MDE1173575.1 MAG4	33.766	5.13e-18
MDE1164404.1 MAG15	MDE1172124.1 MAG4	26.778	1.37e-15
MDE1164404.1 MAG15	MDE1197284.1 MAG14	24.737	2.97e-15
MDE1165299.1 MAG15	MDE1165299.1 MAG15	100.000	0.0
MDE1165299.1 MAG15	MDE1168779.1 MAG15	45.395	3.96e-144
MDE1165299.1 MAG15	MDE1169134.1 MAG15	29.114	6.57e-53
MDE1165299.1 MAG15	MDE1141657.1 MAG6	30.151	1.03e-49
MDE1165299.1 MAG15	MDE1181026.1 MAG7	27.069	1.56e-41
MDE1165299.1 MAG15	MDE1180490.1 MAG7	26.622	1.61e-41
MDE1165299.1 MAG15	MDE1164404.1 MAG15	34.010	5.42e-35
MDE1165299.1 MAG15	MDE1174304.1 MAG4	30.091	4.43e-33
MDE1165299.1 MAG15	MDE1159551.1 MAG8	28.155	1.36e-29
MDE1165299.1 MAG15	MDE1197273.1 MAG14	29.717	4.04e-29
MDE1165299.1 MAG15	MDE1186864.1 MAG9	31.963	4.65e-28
MDE1165299.1 MAG15	MDE1167041.1 MAG15	28.638	1.29e-27
MDE1165299.1 MAG15	MDE1173575.1 MAG4	31.658	4.34e-27
MDE1165299.1 MAG15	MDE1197284.1 MAG14	27.586	5.29e-23
MDE1165299.1 MAG15	MDE1172124.1 MAG4	28.837	9.46e-23
MDE1168779.1 MAG15	MDE1168779.1 MAG15	100.000	0.0
MDE1168779.1 MAG15	MDE1165299.1 MAG15	45.395	3.83e-144
MDE1168779.1 MAG15	MDE1141657.1 MAG6	30.597	2.48e-57
MDE1168779.1 MAG15	MDE1169134.1 MAG15	29.199	1.07e-54
MDE1168779.1 MAG15	MDE1181026.1 MAG7	30.602	3.53e-47
MDE1168779.1 MAG15	MDE1180490.1 MAG7	29.787	1.41e-45
MDE1168779.1 MAG15	MDE1164404.1 MAG15	25.604	2.30e-36
MDE1168779.1 MAG15	MDE1197273.1 MAG14	29.302	1.04e-29
MDE1168779.1 MAG15	MDE1186864.1 MAG9	31.429	4.47e-27
MDE1168779.1 MAG15	MDE1167041.1 MAG15	27.315	1.74e-26
MDE1168779.1 MAG15	MDE1159551.1 MAG8	26.887	1.30e-25
MDE1168779.1 MAG15	MDE1172124.1 MAG4	28.571	4.52e-23
MDE1168779.1 MAG15	MDE1173575.1 MAG4	27.189	5.68e-22
MDE1168779.1 MAG15	MDE1174304.1 MAG4	29.032	8.02e-22
MDE1168779.1 MAG15	MDE1197284.1 MAG14	27.273	8.74e-20
MDE1167041.1 MAG15	MDE1167041.1 MAG15	100.000	0.0
MDE1167041.1 MAG15	MDE1197273.1 MAG14	89.767	0.0
MDE1167041.1 MAG15	MDE1159551.1 MAG8	46.667	1.10e-131
MDE1167041.1 MAG15	MDE1173575.1 MAG4	43.276	3.62e-124
MDE1167041.1 MAG15	MDE1197284.1 MAG14	39.066	7.08e-102
MDE1167041.1 MAG15	MDE1174304.1 MAG4	35.407	8.52e-43
MDE1167041.1 MAG15	MDE1172124.1 MAG4	32.520	1.31e-34
MDE1167041.1 MAG15	MDE1169134.1 MAG15	32.161	9.28e-33
MDE1167041.1 MAG15	MDE1165299.1 MAG15	28.638	6.18e-28
MDE1167041.1 MAG15	MDE1168779.1 MAG15	27.315	1.33e-26
MDE1167041.1 MAG15	MDE1164404.1 MAG15	30.811	6.92e-26
MDE1167041.1 MAG15	MDE1141657.1 MAG6	25.571	8.48e-26
MDE1167041.1 MAG15	MDE1181026.1 MAG7	27.442	2.06e-25
MDE1167041.1 MAG15	MDE1186864.1 MAG9	30.638	2.50e-25
MDE1167041.1 MAG15	MDE1180490.1 MAG7	26.047	2.33e-24
MDE1169134.1 MAG15	MDE1169134.1 MAG15	100.000	0.0
MDE1169134.1 MAG15	MDE1168779.1 MAG15	29.199	1.01e-54
MDE1169134.1 MAG15	MDE1180490.1 MAG7	30.485	1.00e-53
MDE1169134.1 MAG15	MDE1165299.1 MAG15	29.114	3.58e-53
MDE1169134.1 MAG15	MDE1181026.1 MAG7	29.561	4.44e-53
MDE1169134.1 MAG15	MDE1165299.1 MAG15	29.114	5.97e-53
MDE1169134.1 MAG15	MDE1141657.1 MAG6	26.437	1.12e-52
MDE1169134.1 MAG15	MDE1174304.1 MAG4	33.171	4.87e-35

MDE1169134.1 MAG15	MDE1167041.1 MAG15	32.161	9.35e-33
MDE1169134.1 MAG15	MDE1197273.1 MAG14	31.156	1.29e-31
MDE1169134.1 MAG15	MDE1164404.1 MAG15	32.227	2.32e-31
MDE1169134.1 MAG15	MDE1159551.1 MAG8	34.225	8.65e-31
MDE1169134.1 MAG15	MDE1186864.1 MAG9	32.212	1.91e-30
MDE1169134.1 MAG15	MDE1197284.1 MAG14	31.034	3.90e-27
MDE1169134.1 MAG15	MDE1173575.1 MAG4	29.947	3.25e-25
MDE1169134.1 MAG15	MDE1172124.1 MAG4	30.151	4.65e-23

Line 257: CAZyme categories are too broad to make correct assumptions. I suggest to also include an activity-based categorization and see whether some CAZyme groups related with the use of certain bark-polysaccharides are more abundant than others. I suggest to use the Figure S1 of the following paper as guidance for this classification: <https://journals.asm.org/doi/full/10.1128/msystems.00829-22>

Response: good suggestion. We have made two new Supplemental Figures, S16 and S17 that include the expected activities from each CAZyme family, below. Note that for some polyspecific families the activity can still not be accurately predicted, and more generic terms have been used, e.g. β -glycosidase. In Figure S17, we show the corresponding total sum of the predicted abundance of enzymes targeting different carbohydrate categories within the MAGs.

The following text has been added on lines 261-262: “Overall, the abundance of predicted CAZyme activities were highest for microbial glycans and starch, and comparatively low for plant cell wall polymers except for pectin (Figure S16 & S17).”

		MAG														
		10	12	11	3	13	5	1	4	8	6	7	2	9	14	15
Algal/Proteoglycans	Expected activity															
	hyaluronate lyase	PL8 1														
	alginate lyase	PL7_1 1														
	alginate lyase	PL7 1 1														
	alginate lyase	PL6_1 1 1														
	alginate lyase	PL5_1 1 2														
	alginate lyase	PL5 3 1 1 1														
	endo-β-1,4-glucuronan lyase	PL38 1 2														
	chondroitin lyase/chondroitinase	PL35 1														
	alginate lyase	PL34 1														
	hyaluronan,chondroitin sulfate	PL33_1 1 1														
	heparin-sulfate lyase	PL12 2														
	unsaturated glucuronyl hydrolase	GH88 1														
	β-glucuronidase	GH79 4 1 1 1														
	laminarinase	GH55 1 1														
	β-agarase	GH50 1 1 1 1														
	sialidase	GH33 3 1 1 1														
	sialidase	GH30_1 1 1														
	laminarinase	GH17+GT2 2 1 1 2														
	laminarinase	GH17 1 1 1														
exosialidase	GH156 6															
α-N-acetylgalactosaminidase	GH109 1 1 6 2 2 3 1 1 1 3 2															
rhamnogalacturonyl hydrolase	GH105 2 3 1 1															
acetyl esterase	CE9 2 1 3 1 1 1 1 2 1 2 1															
β-agarase	CBM9+GH50 1															
beta-glycosidase	β-galactosidases	GH42 2 2 2 2														
	β-xylosidase	GH39 3 3 1 4 1 3 2														
	β-galactosidases	GH35 2 2 3 3 2 1														
	β-glycosidase	GH3 4 1 3 2 6 2 6 1 1 2 2 5 2 6 5														
	β-glycosidase	GH2 4 3 1 11 4 1 1 1 2														
	β-galactosidase	GH165 1														
	β-glucosidases/β-galactosidases	GH1+GT4 1														
	β-glucosidases/β-galactosidases	GH1 3 1 1 2 1 1 1 1 6														
	β-gal	CBM57+GH2 2 1														
	β-gal	CBM32+GH2 1														
Cellulose	endo-glucanases	GH9 1														
	β-endo	GH8 1 1 1 1 2 1 1 1														
	endo-β-1,4-glucanase	GH5_5 1 2														
	no known	GH5_39 1														
	endo-β-1,4-glucanase	GH5_25 1														
	endo-β-1,4-glucanase	GH5_2 1														
endoglucanase	GH5 2 1 1															
cellulase	CBM2+GH5_1 1															
Chitin	α-N-acetylglucosaminidase	GH89 1														
	N-acetylglucosaminidase	GH73 1 1 2 1 1 1 1														
	chitinase	GH46 1														
	β-N-acetylhexosaminidase	GH20 5 1 3 3 2 3 2														
	chitinase/lysozyme	GH19 2														
	chitinase	GH18 1 3 4 1														
	chitinase/esterase	CE4+GH18+GT2 1 1 1														
	acetyl esterase	CE4 2 2 2 2 1 1 2 1 3 3 2 4 3 2 1														
	N-acetylglucosamine deacetylase	CE14 3 2 2 2														
	chitinase	CBM73+GH18 2														
chitinase	CBM5+GH18 1															
chitinase/lysozyme	CBM5+CBM73+GH19 1															

Starch/Glycogen	α -glycosidase	GH4				1																
	α,α -trehalase	GH37	1	1	1		1	1			1	1	1	2						1		
	α -glucosidase	GH31	2	2	4	1	2	5						2						1		
	glucoamylase	GH15	2	3	2			1			1	1	2	2	1	2	2					
	amylase- α -1,6-glucosidase	GH133						1						1								
	α -glucosidase	GH13+GH13											1	4								
	α -amylase	GH13_7				1																
	trehalose synthase	GH13_33																		1	1	
	α -1,4-glucan/phosphate α -maltosyltransferase	GH13_3	1	1																1	1	
	α -glucosidase	GH13_26	1	1	1	1					1	1	1		1	1	1			1	1	
	α -glucosidase	GH13_23	1	1					1		2											
	sucrose phosphorylase	GH13_18																		1		
	trehalose synthase	GH13_16	1	1	1								1	1						1	1	
	isoamylase	GH13_11+GH77															2					
	α -glucoside linkages	GH13	2	2	3		2	2							2							
	glycogen branching enzyme	CBM48+GH13_9						1	1			1	1	1						1	1	
	isoamylase	CBM48+GH13_11	2	2	1	1			1			2	2	2						1	1	1
	α -amylase	CBM48+GH13_10	1	1	1							1	1	1						1	1	1
	Xylan	xylanase	GH30																		1	
		α -(4-O-methyl)-glucuronidase	GH115																		1	
xylanase		GH10	1																	1		
acetyl xylan esterase		CE6																		1		
acetyl xylan esterase		CE2	1																	1		
4-O-methyl-glucuronoyl methylesterase		CE15	1	1										1								
acetyl xylan esterase		CE1	2	1			3	1	2		2	4	3	1								
endo- β -1,6-glucanase	CBM13+GH30_3	1																				
Xylan/pectin	α -L-arabinofuranosidase	GH51	1	1	5		2						1									
	β -D-galacto/ α -L-arabinofuranosidase	GH5_13				1		1	1													
	α -L-arabinofuranosidase	GH43_9												1								
	β -D-galactofuranosidase/endo- α -1,5-L-arabinanase	GH43_37						1														
	β -D-galacto/ α -L-arabinofuranosidase	GH43_34				1																
	exo- α -1,5-L-arabino/ α -L-arabinofuranosidase	GH43_26				1														1		
	β -D-galacto/ α -L-arabinofuranosidase	GH43_19+GH43_29+GH43_34												1								
	α -L-arabinofuranosidase	GH43_19												1								
	α -L-arabinofuranosidase	GH43_18												1								
	β -1,4-xylosidase	GH43_11																		1		
	α -L-arabinofuranosidase β -D-galactofuranosidase	GH159				1																
	β -L-arabinofuranosidase	GH146	1	1					1							1						
β -L-arabinofuranosidase	GH142	1						1														
β -L-arabinofuranosidase	GH137							1														
Xyloglucan	α -L-fucosidase	GH95	2		1		3													1		
	endo-xyloglucanase	GH74																		1		
	α -xylosidase	GH31	2	2	4	1	2	5											2			
	acetylcysteine	CE20	3		2	2	3													1		

Figure S2. Predicted activities and possible target substrates for relevant CAZy family members identified in the MAGs. Note that some families appear in different categories, and predictions may not show all possible activities, especially from polyspecific families.

Figure S3. Number of CAZyme-encoding genes overall within the MAGs, based on the predicted activities in Figure S16.

Line 345: add DEGRADATION after bark

Response: the article we refer to did not study bark degradation per se, but rather which microbes live on the bark of living trees and how they influence methane release. The methane production is of course likely influenced by some kind of metabolism of bark-related compounds, but to call it bark degradation would be misleading. We have instead changed the sentence to:

Lines 351-353: “The importance of the bacterial communities that exist on bark of living trees was recently highlighted, where it was shown that bark-dwelling methanotrophic bacteria decrease methane emissions from trees”

Line 418: =DSM 114633 is missing a T (of type strain) in the end of the code.

Response: thanks for spotting this. It has been added.

Line 511: “DNA extraction for fungal and bacterial gene regions” as it is, this title does not make sense.

Response: the heading has been changed to “DNA extraction for sequencing of fungal and bacterial gene regions”.

Lines 562-563: SILVA database, release 132 is a bit old version of the database. Currently the 138 version is being used.

Response: the analysis was performed during 2019-2020 and the version used is correctly stated.

Line 597: ribosomal and not ribosomale

Response: this has been corrected.

Line 658: “known species” is super broad! You probably mean known *Pseudomonas* species. In any case, please check validated species in bacterio.net. Also, as I mentioned before, *Pseudomonas* is such an enormous genus that the inclusion of distant species does not make sense in order to evaluate the possibility of *P.abiericivorans* being a new species. You can leave the broad analysis if you want to show the position regarding other “resin acid” (as you name them) *Pseudomonas* species, but for the species description a more reduced tree including just related species (and all of the closest ones) is required.

Response: we agree that was not accurately phrased. As mentioned previously, the phylogenetic tree has been remade (Fig. S20) to include the closest related species based on 16S similarity, and additionally *Pseudomonas* species known to metabolize resin acids. The text has been changed, on lines 706-707, to: “To compare the similarity between *P. abieticivorans* and the most closely related known species, ...”

Figure S4. Phylogenetic tree based on the 16S rRNA gene sequences of *P. abieticivorans* (indicated in orange as *P. sp. PIA 16*) and the ten phylogenetically most closely related members of *Pseudomonas* from NCBI: GenBank, in black. Five species known to metabolize resin acids are shown in green.

Line 604: which version of dbCAN?

Response: dbCAN2 was used and the information has been added.

Description of new species within the genus *Pseudomonas* usually includes the analysis of respiratory quinones.

Response: there are also many papers on new *Pseudomonas* species that do not include this information, so it is not a strict criterion.

Fig S3 is missing the description of the F panel.

Response: this has been corrected.

Reviewer #1 (Remarks to the Author):

The issue with the paper raised in my first review remains. My point was that you have a moisture content so high as to induce anaerobic conditions not representative of nature. The authors have provided a Conference paper as the rebuttal, stating that moisture content was around 70% in this submitted paper and similar to the 60-65% stated in the conference paper (EPJWebofConferences DOI:10.1051). That Conference paper has not given specific methods for moisture content calculation, but wood science moisture content is typically on a dry weight denominator. 60-65% of the weight of bark is not water - it would be like jelly if it were - 60-65% dry weight MC is 37-39% MC as you are calculating based on total weight (water + bark). For 70% MC using wood science rules (dry weight denominator), you would add 0.7 mL of water to 1 gram of bark. Adding 2.3 mL of water to 1 gram of bark = 230% MC by dry weight denominator. It is way way above the fiber saturation point where aerobic conditions transition to anaerobic, which is usually around 80% using the dry weight denominator.

I cannot tell from the photo, just using the numbers - my point remains about anaerobic conditions. Wood science moisture content calculations make little sense to anyone but a kiln drier, but they are the norm, unfortunately.

Reviewer #2 (Remarks to the Author):

Dear Authors,

thank you for your detailed answers. While I am happy with most of them, there are few of them which need further discussion. Please see my replies to previous comments below.

@ Fig S2

Thank you for clarification. So in this way, ash is a proxy to state of the material at the beginning of the experiment. Since its mass is constant, other variables standardized using ash mass show proportions against the original state of the sample/bark. I think it does not need further changes.

@ Fungal amplicon data

I understand that authors would like to present fungal data since fungi show pattern distinct to that of bacteria in the current version of manuscript. However, in order to be able to verify diversity patterns of fungi, added methods paragraph needs improvements. Please clarify if extraction of ITS region and removal of flanking ends was performed as suggested e.g. by Tedersoo et al. 2022. Please note, that this is not equal to trimming.

In order to not under- or over-estimate fungal diversity, I suggest to stick to OTU approach in the case of fungi (as it was performed for bacteria) as suggested by Tedersoo et al 2022 and Estensmo et al. 2021. Please clarify why amplicon sequence variants approach was selected.

Otherwise, I strongly recommend to reanalyse this part of the manuscript utilising ITS extraction + OTU approach.

@ replication description

Thank you for clarification. Based on it, I suggest to avoid using boxplots for duplicated measurements in the main text and supplementary figures as recommended by Krzywinski and Altman (2014).

Please correct Figure 2 caption, median is misleading, lines show means instead (L219)

@ abundance based on metagenome

I feel that metagenomic part needs further clarification. It is difficult to believe in its quantitative conclusions.

Relative abundance of bacterial MAGs can be delineated by mapping of reads to contigs(= genomes). This was done correctly using CoverM.

Such approach is limited in the case of Eukaryota on contig level because of problematic assembly (Saraiva 2023). On read level quantification might work, although the taxonomic resolution of 151bp reads is coarse. Thus, I can not agree with statements on L102, L226-L227 and L400-L402 based on current analysis. Correctness of proportions in Fig S8C is questionable. qPCR or similar quantitative technique would be more suitable to describe bacterial/fungal biomass ratio rather than metagenomics.

My suggestions regarding metagenomics are:

- Please elaborate more why two assemblies were performed (Megahit and Flye) as reason for double assembly might not be clear. Assembly utilizing long and short reads should out-perform short-read-based one.
- I suggest to exclude Eukaryotic analysis from metagenomic part and to use potential of valuable metagenome to describe abundance of MAGs and their genomic potential (that's presented).
- My previous suggestion was to map metagenomic reads against complete genome from *Pseudomonas* PIA16 isolate which might show the most precise abundance (more precise than abundance of fragmented MAG15).
- I suggest to draw conclusions about bacteria vs fungi based on diversity of each group in amplicon datasets instead of metagenome as we do not know more about fungal activity in terms of expression/protein synthesis.

@ dbCAN version

Please add correct version of dbCAN database as it influences the diversity of CAZy families.

Individual versions can be found on dbCAN2 server:

8/9/2022: dbCAN HMMdb v11

8/17/2021: dbCAN HMMdb v10

8/04/2020: dbCAN HMMdb v9

etc.

REFERENCES

Tedersoo, L. et al. Best practices in metabarcoding of fungi: From experimental design to results. *Molecular Ecology* 31, 2769–2795 (2022).

Estensmo, E. L. et al. The influence of intraspecific sequence variation during DNA metabarcoding: A case study of eleven fungal species. *Molecular Ecology Resources* 21, 1141–1148 (2021).

Krzywinski, M., Altman, N. Visualizing samples with box plots. *Nat Methods* 11, 119–120 (2014)

Saraiva, J. P. et al. Recovery of 197 eukaryotic bins reveals major challenges for eukaryote genome reconstruction from terrestrial metagenomes. *Molecular Ecology Resources* 1755–0998.13776 (2023)

Reviewer #3 (Remarks to the Author):

Most of my concerns have been addressed and I believe that the manuscript includes a nice work. Nonetheless, I am still not entirely happy with the fact that both the chemical and the microbial community analyses are performed after the inoculation of an artificial microbial consortium. The authors declare that the questions they try to answer are "how is the bark degraded chemically and which type of microbes may be involved?". If those questions are related to an advance in the knowledge of the ecology of bark degradation, I believe they cannot be solved the way the experiments were set. Regarding the first question, you can see only how the bark is chemically degraded in your artificial conditions. With different (natural) microbial communities, with their specific metabolic pathways, the succession of metabolites during the bark degradation process (the chemistry) are probably different. The same applies for the answer to the microbes

which are involved in bark degradation... we are probably missing important taxa involved in bark degradation. So as the manuscript states, "...bark decomposes in nature, though by which species and mechanisms remains unknown" is still unknown after this work.

The new discussion section has focused on the usefulness of the work from a biotechnological point of view, which I think it is more appropriate in this case. But still, I think some conclusions regarding the applications of the obtained results should be included if this is the sense of the work.

Response to reviewers

We would like to again thank the reviewers for their comments and suggestions, and we were happy to see that most of our previous revisions were appreciated. We have addressed the new comments, point-by-point, and provide answers below each in blue text. We provide both a new clean version of the manuscript, and a marked-up version. In addition to changes based on the reviewers' suggestions, we have also corrected a few typos that were found. Line numbers are indicated where appropriate, and refer to the new version of the manuscript. We believe that our manuscript has been much improved both in terms of clarity and quality, and we hope it is seen as worthy of publication in Nature Communications.

On behalf of all the authors,

Johan Larsbrink

Reviewer #1 (Remarks to the Author):

The issue with the paper raised in my first review remains. My point was that you have a moisture content so high as to induce anaerobic conditions not representative of nature. The authors have provided a Conference paper as the rebuttal, stating that moisture content was around 70% in this submitted paper and similar to the 60-65% stated in the conference paper (EPJWebofConferences DOI:10.1051). That Conference paper has not given specific methods for moisture content calculation, but wood science moisture content is typically on a dry weight denominator. 60-65% of the weight of bark is not water - it would be like jelly if it were - 60-65% dry weight MC is 37-39% MC as you are calculating based on total weight (water + bark). For 70% MC using wood science rules (dry weight denominator), you would add 0.7 mL of water to 1 gram of bark. Adding 2.3 mL of water to 1 gram of bark = 230% MC by dry weight denominator. It is way way above the fiber saturation point where aerobic conditions transition to anaerobic, which is usually around 80% using the dry weight denominator.

I cannot tell from the photo, just using the numbers - my point remains about anaerobic conditions. Wood science moisture content calculations make little sense to anyone but a kiln drier, but they are the norm, unfortunately.

Response: we see the point the reviewer is making and agree that if we use the wood science rules, we will have a much larger moisture content than when basing it on the whole material, and that the term could be misinterpreted. Since we are not studying wood here, even though the bark contains classical wood polymers, we think the best solution is to avoid saying moisture and instead describe it as dry matter content. Moisture has been changed to dry matter overall in the text and the values updated, on lines 477-478.

We listed two examples in our previous response, one conference paper and one peer-reviewed paper, where the conference paper was not cited in the actual manuscripts but just served as an additional example in addition to reference 43 (Kemppainen et al. 2014). In ref 43, the dry matter content was reported for fresh spruce bark as 39% for winter bark and 51% for summer bark, i.e. corresponding to

61-49% wet matter. We have now also added a second peer-reviewed paper as reference (#44, Routa et al. 2020), where the average moisture content of freshly harvested bark between (56.2-62.2%) is reported, and here the moisture was calculated based on the total (wet) mass of the sample (see reference 33 in that paper), i.e. “wet-matter” as in our calculations in this manuscript and not according to the wood science procedure. Reference 44 has been cited on lines 407 and 478.

We have to disagree with the reviewer that our conditions are “not representative of nature” and that our numbers cannot be as high because we would have a jelly—direct comparisons to wood are not applicable for this material. Given the dry/moisture values reported in the two cited references, and our own observations, we hope that the information on bark being quite a moist material naturally, especially compared to wood, is clarified. The values reported in the references are from freshly harvested trees/bark, meaning that such high moisture compared to wood is not an unnatural condition. Once a tree falls on the ground, or if the bark is removed and collected in a large pile as in industry, it would also not be unreasonable that the moisture can be even higher as the bark is no longer on a free-standing tree fully exposed to air. One could then of course speculate that the bark of trees is then an anaerobic tissue in general, but we do not know of any literature referring to bark as intrinsically anaerobic and we do not believe that to be the case. The taxa we identify are not from obligate anaerobes, and we would have expected to see e.g. *Clostridia* or methanogens had that been the case.

Changes to the text:

Discussion, lines 405-407: “..., such micro-environments would likely be minor.” has been changed to “..., such micro-environments would likely not be highly abundant. The dry matter content used in the experiment (30%) was close to that of fresh spruce bark (typically around 35-40%)^{43,44}, which is not known as an anaerobic tissue.”

Methods, lines 477-478: changed moisture to dry matter and including the new reference (#44).

References:

#43. Kempainen et al. 2014. Spruce bark as an industrial source of condensed tannins and non-cellulosic sugars. *Industrial Crops and Products*, 52, 158-168.

<https://doi.org/10.1016/j.indcrop.2013.10.009>

#44 (new). Routa et al. 2020. Effects of storage on dry matter, energy content and amount of extractives in Norway spruce bark. *Biomass and Bioenergy*, 143, 104821.

<https://doi.org/10.1016/j.biombioe.2020.105821>

Reviewer #2 (Remarks to the Author):

Dear Authors,

thank you for your detailed answers. While I am happy with most of them, there are few of them which need further discussion. Please see my replies to previous comments below.

@ Fig S2

Thank you for clarification. So in this way, ash is a proxy to state of the material at the beginning of the experiment. Since its mass is constant, other variables standardized using ash mass show proportions against the original state of the sample/bark. I think it does not need further changes.

Response: thank you for the comment.

@ Fungal amplicon data

I understand that authors would like to present fungal data since fungi show pattern distinct to that of bacteria in the current version of manuscript. However, in order to be able to verify diversity patterns of fungi, added methods paragraph needs improvements. Please clarify if extraction of ITS region and removal of flanking ends was performed as suggested e.g. by Tedersoo et al. 2022. Please note, that this is not equal to trimming.

In order to not under- or over-estimate fungal diversity, I suggest to stick to OTU approach in the case of fungi (as it was performed for bacteria) as suggested by Tedersoo et al 2022 and Estensmo et al. 2021. Please clarify why amplicon sequence variants approach was selected.

Otherwise, I strongly recommend to reanalyse this part of the manuscript utilising ITS extraction + OTU approach.

Response: we agree that using the same approach for both bacteria and fungi is good. We have now re-sequenced all our samples using Oxford Nanopore Technology (ONT) and primers bV18-A for bacteria and fITS2-C for fungi and this more up-to-date method gives a higher taxonomic resolution and thereby is able to capture more of the sequence diversity than short-read sequencing, and we have now used an OTU approach for both datasets. The sequencing rarefaction curves indicate that we are well above the sequencing depth needed for the analyses, as can be seen below this response. Compared to the previous short-read amplicon data, for the fungi we see highly similar trends over the experiment, though with a somewhat higher abundance of Leotiomycetes at the later stages of the experiment. The Shannon diversity was lower than before (especially for timepoint zero, resulting in less of a sharp initial drop). For bacteria, the Shannon diversity is instead higher throughout the experiment but otherwise appears highly similar to the previous short-read results, like for the fungi. Note that the reads were mapped against the SILVA 16S/18S rRNA 138 SSURef NR99 full-length database, and some recent taxonomic changes have resulted in certain re-classifications, such as Betaproteobacteriales (formerly Betaproteobacteria) is now Burkholderiales, an order of Gammaproteobacteria.

No removal of flanking 18S and 28S rRNA genes was carried out. However, we matched our data against the Unite database and only sequences with alignment length >125 bp and mapping quality >75% were considered for further analysis. As evident from the mapping quality and length histograms, below, most sequences mapped with above 75% mapping quality with alignments lengths >500 bp.

The methods section has been updated accordingly, Library preparation and sequencing of bacterial and fungal marker gene regions on lines 568-592, Bioinformatic processing bacterial gene regions on lines 593-606, and Bioinformatic processing fungal gene regions on lines 607-617. The sequencing statistics have been updated in Table S3 and PCA in Figure S6 and alpha-diversity in Figure S7.

Rarefaction curves:

Histograms:

@ replication description

Thank you for clarification. Based on it, I suggest to avoid using boxplots for duplicated measurements in the main text and supplementary figures as recommended by Krzywinski and Altman (2014).

Please correct Figure 2 caption, median is misleading, lines show means instead (L219)

Response: we agree and have removed the boxplots, and thank you for spotting the caption error for Figure 2 which has been corrected.

@ abundance based on metagenome

I feel that metagenomic part needs further clarification. It is difficult to believe in its quantitative conclusions.

Relative abundance of bacterial MAGs can be delineated by mapping of reads to contigs(= genomes). This was done correctly using CoverM.

Such approach is limited in the case of Eukaryota on contig level because of problematic assembly (Saraiva 2023). On read level quantification might work, although the taxonomic resolution of 151bp reads is coarse. Thus, I can not agree with statements on L102, L226-L227 and L400-L402 based on current analysis. Correctness of proportions in Fig S8C is questionable. qPCR or similar quantitative technique would be more suitable to describe bacterial/fungal biomass ratio rather than metagenomics.

Response: we agree with the reviewer that this can be explained better. While the data underlying Figure S8C provides insight with respect to the final metagenome assembly, a read-based approach should be pursued to get the assembly-unbiased taxonomic resolution (only limited by the database). qPCR is another approach but with recent advances in databases, we consider it less ideal and this approach is also subject to other biases, such as primer/PCR reaction biases. ONT data provides much fewer unclassified reads than Illumina (longer reads = more taxonomic information per read) and our data were classified against the Kaiju non-redundant database + euk database. The recent (updated) data (shown in Figure S8) employs a database with 321 million protein sequences, which should be adequate/trusted for e.g. Phylum-level classification.

Please also see response below regarding the abundance/roles of fungi within the microbial enrichment.

My suggestions regarding metagenomics are:

- Please elaborate more why two assemblies were performed (Megahit and Flye) as reason for double assembly might not be clear. Assembly utilizing long and short reads should out-perform short-read-based one.

Response: this project was performed incrementally over several years, first using Illumina and then with additional ONT long-read data to improve quality and if possible capture important and high-quality MAGs such as our isolated *Pseudomonas* MAG, and not necessarily to describe the entire metagenome. The Illumina data was used to probe the composition and, if possible, capture the *Pseudomonas* PIA16 (which was not entirely possible without the ONT data). We have clarified in the text that the two methods were done sequentially and the reasons for this, on lines 224-227: “To gain deeper biological insight on the process, short-read metagenome sequencing was first performed on the 2-week sample, and later followed by long-read sequencing to improve quality and enable reconstruction of metagenome-assembled genomes (MAGs) (Table S4 and S5).”

- I suggest to exclude Eukaryotic analysis from metagenomic part and to use potential of valuable metagenome to describe abundance of MAGs and their genomic potential (that's presented).

Response: we agree that we should not overinterpret the eukaryote abundance from the metagenomes, and a complementary technique could have shed light on this matter. Ideally, we would have used

metaproteomics to get a better understanding of which enzymes are abundant and used during the bark degradation, as even if fungi are under-represented on the DNA level, they could still perform key functions, but unfortunately the storage of our samples makes such follow-up analyses impossible (i.e. not stored at -80°C). We still believe reporting the values of abundances of reads and contigs can be relevant information, especially for comparative future studies for example to investigate the correlation of reads/contigs vs biological roles, but we have now refrained from drawing further conclusions from these. We have clarified that these numbers should not be overinterpreted and that in future studies also other complementary techniques should ideally be used to infer biological roles. Changes to the text are:

Lines 227-229, to report only numbers: “Within the metagenome, a majority of reads from the long-read sequencing were classified as bacterial (91%), with lower numbers for eukaryotes (3%), and the remaining reads classified as archaeal, viral, or unclassified reads (Fig. S8).”

Line 390: removed “more dominant” referring to bacteria.

Lines 398-401: “Without complementary techniques, such as quantitative metaproteomics, the biological roles of the fungi remain more elusive as the number of reads from sequencing does not necessarily reflect biological importance, and fungi are often regarded as main drivers of boreal forest biomass conversion.”

- My previous suggestion was to map metagenomic reads against complete genome from *Pseudomonas* PIA16 isolate which might show the most precise abundance (more precise than abundance of fragmented MAG15).

Response: while fragmented, MAG15 is still considered a MIMAG high-quality MAG. Cultivation can also lead to domestication but we do not find this too likely with the short time frame from isolation to sequencing. That said, we have re-mapped the data using CoverM. We have used both default (75% sequence identity) and more stringent (99% identity) mapping settings, which shows that with default settings, PIA16 abundance in the 2-week enrichment metagenome is 34.1%, and with stringent settings which should in theory only map exactly the PIA16 strain, it is 1.24%. The interpretation of this is that the population represented by the PIA16/MAG15 genomes is a collective of very closely related strains (i.e. microdiversity), of which PIA16 is one, and we can assume that they share metabolic capabilities. We believe this is an interesting observation that suggests that many other strains of *P. abieticivorans* remain to be discovered, and have updated the discussion to point towards this possibility. The following changes have been done:

To indicate that PIA16 is one of several species, we have updated the end of the Introduction, on Lines 96-98: “Bacteria rather than fungi were linked to resin acid degradation, and we successfully isolated and genome-sequenced a strain of the main resin acid-degrading bacterium, which represents the new species *Pseudomonas abieticivorans*.”

And on Lines 293-297: “Coverage estimations of the complete PIA16 genome in our resin-degrading enrichments (2-week sample) showed it to be 34.1% of the total metagenome, with CoverM default settings, and 1.24% with more stringent settings (99% identity), suggesting that the PIA16/MAG15 population likely represents a collection of highly similar strains exhibiting microdiversity but presumably sharing metabolic capabilities.”

- I suggest to draw conclusions about bacteria vs fungi based on diversity of each group in amplicon datasets instead of metagenome as we do not know more about fungal activity in terms of expression/protein synthesis.

Response: please see previous responses.

@ dbCAN version

Please add correct version of dbCAN database as it influences the diversity of CAZy families. Individual versions can be found on dbCAN2 server:

8/9/2022: dbCAN HMMdb v11

8/17/2021: dbCAN HMMdb v10

8/04/2020: dbCAN HMMdb v9

etc.

Response: the correct version has been updated on line 662 (version 10).

REFERENCES

Tedersoo, L. et al. Best practices in metabarcoding of fungi: From experimental design to results. *Molecular Ecology* 31, 2769–2795 (2022).

Estensmo, E. L. et al. The influence of intraspecific sequence variation during DNA metabarcoding: A case study of eleven fungal species. *Molecular Ecology Resources* 21, 1141–1148 (2021).

Krzywinski, M., Altman, N. Visualizing samples with box plots. *Nat Methods* 11, 119–120 (2014)

Saraiva, J. P. et al. Recovery of 197 eukaryotic bins reveals major challenges for eukaryote genome reconstruction from terrestrial metagenomes. *Molecular Ecology Resources* 1755–0998.13776 (2023)

Reviewer #3 (Remarks to the Author):

Most of my concerns have been addressed and I believe that the manuscript includes a nice work. Nonetheless, I am still not entirely happy with the fact that both the chemical and the microbial community analyses are performed after the inoculation of an artificial microbial consortium.

The authors declare that the questions they try to answer are "how is the bark degraded chemically and which type of microbes may be involved?". If those questions are related to an advance in the knowledge of the ecology of bark degradation, I believe they cannot be solved the way the

experiments were set. Regarding the first question, you can see only how the bark is chemically degraded in your artificial conditions. With different (natural) microbial communities, with their specific metabolic pathways, the succession of metabolites during the bark degradation process (the chemistry) are probably different. The same applies for the answer to the microbes which are involved in bark degradation... we are probably missing important taxa involved in bark degradation. So as the manuscript states, "...bark decomposes in nature, though by which species and mechanisms remains unknown" is still unknown after this work.

The new discussion section has focused on the usefulness of the work from a biotechnological point of view, which I think it is more appropriate in this case. But still, I think some conclusions regarding the applications of the obtained results should be included if this is the sense of the work.

Response: thank you for the nice words about our work. We see the point the reviewer is making about the consortium, but do not fully agree that we have utilized an "artificial consortium" which suggests that it was random or designed. As we have described, we sourced the consortium from actual spruce bark, coming from an industrial bark pile from de-barking multiple spruce trees, where one can assume that naturally occurring bark-dwelling or bark-decomposing species are found. As we have added to our manuscript text based on the reviewer's last suggestion (lines 393-398), we see similar microbial profiles as have been observed previously on spruce bark-containing samples (Hagge et al. 2019). As we also mention, and which can be inferred from previous studies, bark degradation is very slow, and while our initial enrichment step to validate growth before we inoculated sterile bark most likely affected the inoculum proportions, we detected several hundred OTUs during the experiment indicating that a large part of this consortium did not become extinct. The comparison with previous literature, however limited, and the successful growth of many microorganisms on this complex biomass should suggest that our enrichment contains microbial populations and metabolisms that originate from a naturally occurring community with bark-degrading capacity. Certainly, different compositions of communities will exist in nature and not all be equal to ours, but our study still describes how one complex community sourced from actual bark can in fact be used to re-inoculate and degrade bark.

To say that "in nature", the metabolic pathways and which species are involved are probably not the same as seen here we would say is speculation at this point, and the statement that knowledge on the natural process is still unknown after this work we find a bit harsh, especially when we specified in the text that "In order to understand how bark can be degraded", line 88, and "Our holistic study paves the way for detailed understanding of microbial degradation of bark", lines 100-101. What is meant by natural/nature is subjective, as even any sampling in a forest could be considered artificial based on which forest, soil, part of the tree, season, interference from animals, etc. – the diversity of parameters is simply too high. We do not claim that we have the complete answer to how bark is degraded in nature, and possibly there is not one general process but rather many different ones, but believe we have here laid a very important foundation for understanding how bark can be degraded by microorganisms. Future studies will enable comparative analyses to ours, and it will be highly interesting to see whether those either replicate our results or show differences. To better acknowledge the limitations of the study we have revised the manuscript as follows:

Line 89-90: "what type of species are involved" changed to "what type of species may be involved".

Lines 145-146: added "by the consortium" to this sentence: "..., indicating that bark extractives are a major carbon source during initial metabolism of spruce bark by the consortium."

Discussion, lines 355-356: “Despite its fundamental protective role, how bark can be degraded by microorganisms is not known.”,

lines 401-403: “Follow-up studies of bark-degrading microbiomes will be crucial to determine how diverse such communities can be, and whether our results reflect general trends for microbial spruce bark metabolism.”,

and lines 429-431: “Understanding how bark can be degraded in nature by different microorganisms is key for such future valorization, for instance using select enzymes” has been changed to “Understanding how bark can be degraded by different microorganisms is key for such future valorization, for instance using select enzymes acting on valuable extractives”,

and finally on lines 431-434 we have added a new sentence to emphasize some biotechnological new possibilities that could result from our work: “Our isolated new *Pseudomonas* species, or newly discovered strains thereof, may for instance become a useful source of especially resin acid-modifying enzymes to selectively introduce new functional groups on these interesting molecules.”

Reviewer #2 (Remarks to the Author):

Dear Authors,
while I am satisfied with most of responses, I would like to further comment on amplicon sequencing. I agree that long-reads can provide certain benefits regarding e.g. taxonomic resolution. From that reason, could you please specify mean sequence count \pm SE instead of sequence number range (lines 186-187 marked-up version) together with mean length \pm SE for bacterial and fungal amplicons? Without this, it is difficult to asses quality of alignment length which was displayed in histograms.

Also, please correct line 501 marked-up version: "somewhat below bark sampled..."

Response to reviewer

Answers are provided below, in blue text.

Johan Larsbrink

Reviewer #2 (Remarks to the Author):

Dear Authors,

while I am satisfied with most of responses, I would like to further comment on amplicon sequencing. I agree that long-reads can provide certain benefits regarding e.g. taxonomic resolution. From that reason, could you please specify mean sequence count \pm SE instead of sequence number range (lines 186-187 marked-up version) together with mean length \pm SE for bacterial and fungal amplicons? Without this, it is difficult to asses quality of alignment length which was displayed in histograms.

Thank you for the positive assessment. We have revised the manuscript according to the suggestion, and replaced the sequence number range with mean sequence counts and the mean lengths \$\pm\$ SE, on lines 155-157. The text now reads:

“After quality control and bioinformatic processing, an average of \$64,163 \pm 4,612\$ fungal reads and \$87,290 \pm 4962\$ bacterial reads were generated per sample (Table S3), with mean sequence lengths of \$495 \pm 21\$ and \$1359 \pm 11\$ bp, respectively.”

Also, please correct line 501 marked-up version: "somewhat below bark sampled..."

Thank you for pointing out this error, which has now been corrected.